# Systematic Assessment of Burst Impurity in Confocal-Based Single-Molecule Fluorescence Detection Using Brownian Motion Simulations

**DOI:** 10.3390/molecules24142557

**Published:** 2019-07-13

**Authors:** Dolev Hagai, Eitan Lerner

**Affiliations:** Department of Biological Chemistry, The Alexander Silberman Institute of Life Sciences, Faculty of Mathematics & Science, The Edmond J. Safra Campus, The Hebrew University of Jerusalem, Jerusalem 9190401, Israel

**Keywords:** single-molecule, fluorescence, burst, photon rate, effective detection volume, point-spread function, Brownian, diffusion, simulation, threshold

## Abstract

Single-molecule fluorescence detection (SMFD) experiments are useful in distinguishing sub-populations of molecular species when measuring heterogeneous samples. One experimental platform for SMFD is based on a confocal microscope, where molecules randomly traverse an effective detection volume. The non-uniformity of the excitation profile and the random nature of Brownian motion, produce fluctuating fluorescence signals. For these signals to be distinguished from the background, burst analysis is frequently used. Yet, the relation between the results of burst analyses and the underlying information of the diffusing molecules is still obscure and requires systematic assessment. In this work we performed three-dimensional Brownian motion simulations of SMFD, and tested the positions at which molecules emitted photons that passed the burst analysis criteria for different values of burst analysis parameters. The results of this work verify which of the burst analysis parameters and experimental conditions influence both the position of molecules in space when fluorescence is detected and taken into account, and whether these bursts of photons arise purely from single molecules, or not entirely. Finally, we show, as an example, the effect of bursts that are not purely from a single molecule on the accuracy in single-molecule Förster resonance energy transfer measurements.

## 1. Introduction

The outstanding capability to detect fluorescence from single molecules, one molecule at a time, allowed distinguishing between different molecular species based on many parameters derived from these experiments [1]. In the field of single-molecule fluorescence detection (SMFD) there are two main experimental platforms: i) Fluorescence imaging of single molecules immobilized to the surface of a coverslip; and ii) detection of fluorescence bursts from freely-diffusing single molecules through a small effective detection volume (EDV). The advantage of immobilized SMFD over its freely-diffusing counterpart is that the fluorescence from each individual molecule can be recorded for long periods of time. Still, the main limitation of immobilized SMFD is the inherent requirement to immobilize the molecule in the first place. Within this perspective, the common immobilization is through chemical conjugation, which always raises criticism about the possible perturbation this procedure might cause to the biomolecule under measurement. There are other techniques that allow limiting a biomolecule to a small volume, such as the anti-Brownian electro-kinetic (ABEL) trap [2], or confinement by encapsulation inside vesicles [3,4] or liposomes [5]. Nevertheless, each such method presents its own possible limitations (the electric field used in the ABEL trap to center back the molecules might affect them; it might be hard to exchange chemical conditions if the molecules are encapsulated in liposomes). For the abovementioned reasons, some researchers choose the confocal-based SMFD of freely diffusing molecules, as a simple, yet powerful approach. Still, the non-uniformity of the EDV at different positions in space and the random Brownian motion of molecules into these different spatial regions, make identifying and distinguishing signals of single molecules difficult.

A confocal-based SMFD measurement of freely diffusing molecules involves laser excitation tightly focused through a high numerical aperture objective lens (usually water immersion), where the focus is brought deep (a few tens of μm) inside the sample solution. Then, if the overall laser power is low, effective excitation occurs mostly when molecules traverse through an EDV with a ~1 femtoliter volume. Lenses and a narrow pinhole (diameter of a few tens of μm) focus fluorescence photons, collected by the objective lens, so that only the maximum of focused fluorescence crosses it, while its periphery is rejected. Then, fluorescence is re-collimated by additional lenses, spectrally selected (with proper filters) and focused, so that light reaches the active detection area of single-photon avalanche diodes (SPADs; the detectors). Confocal-based SMFD measurements of freely diffusing molecules is carried out, assuming the concentration of the diffusing molecules is so low (<100 pM) that most of the time no molecule traverses the EDV, and when some do, they are mostly single ones. The measured trace is comprised of a long list of photon detection times (i.e., photon timestamps) relative to the moment the recording started, as well as tagging by the identity of the detection channel, usually used in SMFD applications, in which more than one SPAD is used (such as in single-molecule Förster resonance energy transfer; smFRET). Overall, most of the measured trace consists of background (BG), and there are short intervals of time, in which fluorescence photons from molecules that traversed the EDV are detected. Therefore, SMFD of freely-diffusing molecules require low BG and high fluorescence signals. Aside from choosing fluorophores with high *brightness* (i.e., high absorption coefficient, high fluorescence quantum yield, high photo-stability, and minimum of transitions to dark states) and working with detectors and optics that minimize BG, the way signals from molecules are identified as photon bursts is also of importance. The distinction between BG and signal is found in photon detection rates. If the time interval of all pairs of consecutive photon timestamps (detection moments) will be collected into a histogram, one will identify it includes two time-dependent processes—a slow Poisson process and a fast Poisson (and sometimes super-Poisson) process. If so, identifying fluorescence from molecules that traversed the EDV is a matter of identifying signals with instantaneous photon detection rates high enough relative to the mean rate of the slow Poisson process of the BG.

This procedure is precisely the one used in burst identification in SMFD of freely-diffusing molecules. The single-molecule burst analysis procedure of SMFD has been discussed in many previous works [6,7,8,9,10,11]. The procedure includes a window of *m* consecutive photon detections, for which we calculate the time, *t*, from the first detection time (No. 1) to the last one (No. *m*) [6,12], or the rate, *f*, by dividing *m* by this time interval [11]. In the analysis, this window slides one photon timestamp at a time. These calculations are used for the definition of the signal as photon bursts with instantaneous photon detection rates higher than a given threshold. Doing so, the instantaneous time, *t*, for the *m* consecutive detection events should be smaller than some arbitrary time *T* [6,12]. Using the *m*-rate notation, the rate, *f*, should be higher than some arbitrary rate [11]. To make the threshold choices a bit less arbitrary, one can choose a minimal rate threshold, *F* times higher than the BG rate. This way, at least, the signal is defined relative to the background objectively. Still, the choice of the value of this *F* parameter is arbitrary.

After identification of photon bursts, each burst is defined by the total amount of photons in it (the burst size), the time interval from the first to the last photon in the burst (the burst width), the ratio of the burst size and width (the mean photon brightness), and the maximal instantaneous photon rate (also equivalent to the *brightness*). After burst identification, it is customary to filter photon bursts according to some of these criteria or ratios of them, using, again, some arbitrarily chosen values. Many choose the burst size as a burst selection criterion, where the larger the burst size is, the more trustworthy it is in calculations using the burst size, such as in the calculations of the mean FRET efficiency using burst sizes recorded from detections of two SPADs. However, since in SMFD, conclusions are drawn from burst-dependent histograms (e.g., FRET histograms, burst width histograms), many bursts are required to pass the burst selection criterion, but the higher the minimal burst size threshold is, the lower the amount of bursts that will be selected.

Overall, there is a price for arbitrary choice of parameter thresholds in this process, which is why it is important to understand the meaning of choosing different threshold values at the level of the chosen molecules and their positions in space, when they emitted a photon that was selected as part of a burst. In this work we used PyBroMo (Python Brownian motion) simulations [13] to simulate SMFD of freely-diffusing molecules. Doing so, we recorded both the molecular positions in the simulation and the photon timestamps. Then, after employing the burst analysis procedure, as we usually do in experiments, we assessed the positions of single molecules, at the moments in which each photon of each burst was detected. By doing so, it was possible to define the spatial maps of positions as a function of the burst analysis parameters. We tested different *m*-photon windows, different *F* photon rate thresholds, different burst size, *sz*, and different burst width minimal thresholds. Additionally, we repeated the assessment for different simulations results with different diffusion coefficients at a given molecular concentration, as well as different concentrations at a given diffusion coefficient. We show that in order to identify bursts of molecules that traversed a well-defined narrow region of the EDV, where the bursts are constructed mostly of photons of single molecules, the best practice would be to use large values of photon detection rate thresholds, *F* and modest values of the burst size threshold. We also show that measuring lower concentration of molecules further improves the single-molecule burst identification.

## 2. Results

We used the Python Brownian motion (PyBroMo) [13] code to simulate single-molecule fluorescence detection (SMFD) measurements of molecules freely diffusing in three dimensions (3D) and detected in a confocal-based effective detection volume (EDV). For achieving this purpose, we then simulated photon timestamps using two different PSF models: i) A numerically-calculated PSF model, to mimic a realistic PSF shape, and ii) a Gaussian-shaped PSF model (See shapes of PSF models in Figure 1; for further details, see Materials and Methods). Then we utilized the FRETbursts [14] Python code to analyze the results of the simulations, as if they were actual SMFD experimental results.

The burst analysis parameters that were tested and their values were: i) The number of consecutive photons, *m*, in the burst search sliding window (values 5, 10, 15, and 20, at a constant photon rate threshold, *F* = 6); ii) the photon rate threshold, *F*, defined as the minimal multiplier of the BG rate that can be considered a signal (values 3, 6, 11, 16, and 21, at a constant *m* = 10 consecutive photons); iii) minimal burst size thresholds, *sz* (values 10, 20, 40, and 80, for constant *m* = 10 and *F* = 6; burst size threshold value of 10 is already included for *m* = 10 and *F* = 6, since by definition the identified bursts include at least *m* = 10 consecutive photons); and iv) minimal burst width thresholds, *w* (values 0.0, 0.5 and 1.0 ms, for constant *m* = 10 and *F* = 6; burst width threshold value of 0.0 ms is already included for *m* = 10 and *F* = 6, since by definition the identified bursts include all bursts, with all burst widths).

Appendix A summarizes the different simulation conditions we tested. For further details on the simulation and its analyses, please see Materials and Methods.

### 2.1. Molecular Position Dispersion

The simulations allowed recording the position of each molecule at any instance of the diffusion simulation. Therefore, we plotted the positions of the molecules when they emitted a photon that was detected and was part of a photon burst using the burst analysis parameters shown above. This produced a 3D scatter plot of positions, where we chose to show the two-dimensional (2D) scatter plots of the *xz*, *yz,* and *xy* projections. Next, we overlaid contour lines of the PSF in these 2D projections, for reference (Figure 2). These results are for the simulation molecules in a concentration of 62 pM, where the diffusion coefficient of the molecules was 90 μm^2^/s, using the numerical PSF model. The results for the same simulations using the Gaussian PSF model are summarized in Appendix A.

Figure 2 shows the positions of these molecules at the moment they emitted a photon, after they have been selected as belonging to bursts, either by minimal burst analysis criteria (left) or stringent criteria (right). While minimal criteria were achieved using a sliding window of *m* = 5 consecutive photons, and identifying bursts where the instantaneous photon rate was at least *F* = 6 times higher than the BG rate, stringent criteria were achieved with *m* = 10, *F* = 6 and after selecting bursts with sizes larger than a threshold of *sz* = 40.

In both cases, the positions of the molecules are well within the PSF with a tendency towards its center. However, while in the case of minimal burst analysis criteria the molecular position dispersion (i.e., the spread or the dispersion of the positions) was quite wide, in the case of stringent burst analysis criteria, the molecular position dispersion became smaller. How should one decide which criteria are considered stringent and which not? One quantification to follow would be how well burst analysis parameter values decrease the probability of observing bursts arising from more than single molecules. We term such bursts, impure bursts. We argue that the probability of acquiring impure bursts correlates with the shape of part of the EDV that is explored by molecules that pass the burst search and selection criteria. In simple words, we hypothesize that the larger the molecular position dispersion is, the higher the chances of including bursts of photons not originating solely from single molecules. As a first step, we assess the dependence of the molecular position dispersion in *x*, *y,* and *z*, as a function of different burst analysis criteria and as a function of concentration, diffusion coefficient, simulation time, and PSF models (see Materials and Methods).

Appendix A shows the 1D histograms of the molecular positions as a function of the *z*, *x,* and *y* coordinate (Figure 3, left, center, and right panels, respectively), for the four different tests of the burst search parameter values explained above (from top to bottom, different *m* values with a constant *F*, different *F* values with a constant *m*, different minimal burst size, *sz*, and different minimal burst width, *w*, thresholds with constant *m* and F).

Qualitatively, increasing the different burst analysis parameter values makes the position dispersion smaller, however to different degrees. It is clear, for instance, that increasing the instantaneous photon rate threshold, *F*, and the minimal burst size threshold, *sz*, have the largest effect on decreasing the position dispersion. It is also clear that the minimal burst width threshold, *w*, does not have a significant effect on the molecules’ position dispersion.

To quantitatively assess the effect of choice of burst search parameter values on the molecular position dispersion, we quantified the position dispersion in each dimension by calculating the standard deviations in the *x*, *y*, and *z* coordinates. Figure 3 and Figure 4 report the results of quantifying the molecules’ position dispersion as a function of the burst analysis parameter values assessed in the histograms of Appendix A (concentration of 62 pM and diffusion coefficient of 90 μm^2^/s), as well as for the simulation results in lower concentrations (31 and 15.5 pM) at a constant diffusion coefficient value (90 μm^2^/s; see Figure 3), and in lower diffusion coefficient values (22.5 and 5.625 μm^2^/s) at a constant concentration (62 pM; see Figure 4), using the numerical PSF model. The results using the Gaussian PSF model are also shown in Appendix A.

Among all factors assessed here, increasing the instantaneous photon rate and minimal burst size thresholds, *F* and *sz*, respectively, decreases the molecular position dispersion the most. Additionally, decreasing either the concentration or the diffusion coefficient has a minimal effect on the molecular position dispersion in the z coordinate (in within error ranges) and a larger effect on the molecular position dispersion in the x coordinate (the effect was significant in the z coordinate and minimal in the x coordinate, when using the Gaussian PSF model; see Appendix A). It is noteworthy that the effect of decreasing molecular position dispersion with decreasing concentrations or diffusion coefficients per given burst analysis parameter values, were not always monotonous, sometimes in within the error ranges, but sometimes also outside it.

Next, we assessed the level of burst impurity as a function of different burst search criteria and simulation conditions.

### 2.2. Pure and Impure Single-Molecule Bursts

Burst analysis of SMFD measurements are based on the assumption that photons in each burst are emitted from a single molecule, and hence can be considered *pure single-molecule bursts*. However, it is certainly possible that there will be a fraction of single molecules that will include photons that arise mostly from a single molecule, whereas a few photons from another molecule that crossed the effective detection volume at the same time may be part of the burst. One can imagine a possibility that two molecules traverse the EDV at the same time. However, in the majority of cases, one of the two molecules will be closer to the center of the EDV, hence emitting photons at a higher rate than the other. The overall photon emission rate will still be the combined photon rate, which is why such bursts might be selected and not ignored by the analysis. We term such bursts, *impure single-molecule bursts*.

Accordingly, three factors may influence the occurrence of such bursts and their level of impurity: 1) The choice of the burst search parameter values; 2) the concentration; and 3) the diffusion coefficient. While the first factor is intrinsically controlled in the data analysis, the second and third factors are extrinsic parameters that can be controlled in the experiments.

To assess burst impurity, we identified bursts with photons associated with more than one molecule per burst. Additionally, we calculated the impurity of each burst as the complement of the ratio of the amount of photons arising from the most frequent molecule and the overall amount of photons (as in Equation 1):*f = 1 – (photons from most frequent molecule)/(all photons)*(1)

BG timestamps were discarded from the calculation of the amount of photons. Appendix A shows the histograms of bursts as a function of the level of impurity, expressed as the fraction of photons from other molecules (as in Equation 1).

To quantitatively assess the dependence of burst impurity in different burst analysis criteria and in different simulation conditions, we calculated three mean values, out of histograms as the one shown in Appendix A (for one simulation), over all simulation conditions: The fraction of impure bursts, the mean burst impurity level and the overall mean impurity level (for further details, see Materials and Methods). Figure 5 and Figure 6 show the values of these mean quantities as the fraction of impure bursts (left), and the fraction of impure photons (right), whether as a mean in all bursts (black), or as a value relative to all photons of all bursts (red), respectively. Figure 5 and Figure 6 also report the values of these mean quantities for different burst search criteria as a function of different concentrations, for molecules diffusing with a constant diffusion coefficient (90 μm^2^/s; see Figure 5), and as a function of different diffusion coefficients, at a constant concentration (62 pM; see Appendix A), using the numerical PSF model. These results using the Gaussian PSF model are shown in Appendix A.

For given conditions (concentration and diffusion coefficient), increasing the value of the burst search parameter *m*, increases both the occurrence of impure bursts and their level of impurity. Increasing the value of the burst search parameter *F*, however, decreases burst impurity. Additionally, increasing the value of the burst selection parameter, the burst size threshold *sz*, helps select more impure bursts with higher impurity levels. Changing the value of another burst selection parameter, the burst width threshold, does not influence the burst impurity occurrence and level. Finally, and as expected, the lower the concentration is, the lower are both the occurrence of impure bursts and their level of impurity. Interestingly, decreasing the diffusion coefficient also decreases burst impurity.

In Figure 2, Figure 3 and Figure 4 and Appendix A we have shown that by controlling the burst analysis parameter values, we can modify the molecular positions selected as photons in bursts. Additionally, in Figure 5 and Appendix A we have shown that by controlling the burst analysis parameter values, we can also modify the occurrence of impure bursts and their level of impurity. However, we have also assumed that the two are related. Figure 6 shows both the molecular position dispersion (in the z coordinate) and the bursts’ mean impurity levels, over the different burst search criteria, for different simulations, as a function of different concentrations, for molecules diffusing with a constant diffusion coefficient (90 μm^2^/s), and as a function of different diffusion coefficients, at a constant concentration (62 pM), both when using the numerical and the Gaussian PSF models.

The results shown in Figure 6 clearly show the correlation between burst impurity and the molecular position dispersion: increasing the value of the burst search parameter *m*, leads to a decrease in the molecular position dispersion on one hand, but also to an increase in burst impurity on the other hand. Similar trends are shown for the burst size threshold *sz* and to a lesser extent, also the burst width threshold, *w*. On the other hand, increasing the value of the instantaneous photon rate threshold *F*, leads to a decrease in both the molecular position dispersion and in burst impurity. Additionally, decreasing the concentration leads to a decrease in the dependence of burst impurity on molecular position dispersion.

Regarding the decrease in the values of the diffusion coefficients, it is interesting to mention two main differences that appear when using different PSF models: 1) For increasing values of the instantaneous photon rate threshold *F*, while with a Gaussian PSF model, the correlation between burst impurity and the molecular position dispersion decreases with decreasing diffusion coefficient values, with a numerical PSF model, that correlation does not change; 2) for increasing values of the *m* burst search parameter or the burst size threshold *sz*, while with a Gaussian PSF model, there is an negative correlation between burst impurity and the molecular position dispersion, with a numerical PSF model, the negative correlation at high diffusion coefficient values turns into a positive correlation at low diffusion coefficient values.

While the correlations shown above are apparent correlations between molecular positions and burst impurity, these results do not report on the positions of molecules that contributed impure photons. Figure 7 reports on the molecular positions of photons identified as impure (red), of all photons of impure bursts (yellow), of all photons of all pure bursts (green) and of all photons of all bursts (black), as a function of varying instantaneous photon rate threshold, *F*, values (for the simulation results in concentration of 62 pM and diffusion coefficient of 90 μm^2^/s, using the numerical PSF model).

Qualitatively inspecting the shape of the molecular position histogram of impure photons, relative to the molecular position histogram of photons from pure bursts, it is clear that burst impurity is less pronounced in the center of the EDV, and more pronounced on its rim, at positions where photons of pure bursts are less represented. Nevertheless, there are still many occurrences of impure photons from the center of the EDV. Figure 7 emphasizes the decreasing effect of burst impurity with increasing values of the burst search *F* parameter, as was already shown above. Additionally, increasing the values of the burst search parameter, *m*, or the burst size threshold, *sz*, enhance burst impurity (see Appendix A, respectively), and the shapes of the molecular position histograms of the impure photons stay as reported in Figure 7. Similar trends were also found when performing the simulations using the Gaussian PSF model (Appendix A), only more symmetric than the ones achieved using the numerical PSF model.

Therefore, we conclude that photons arising from molecules other than the main ones in bursts, hence impure photons, have a tendency to originate from molecules on the periphery of the EDV, and that molecules emit photons at a higher rate s they get closer to the EDV center. Therefore, identification of bursts with higher instantaneous photon rate thresholds will include a higher amount of photons originating from the main molecule, and burst impurity thus decreases.

### 2.3. Alternative Quantification of Burst Impurity using Burst Photon Timestamp Autocorrelation

In the previous section, we assessed burst impurity directly from the identity of the molecules that emitted each photon in bursts. Doing so, we quantified the occurrence of burst impurity and the mean level of burst impurity directly by knowing the ground-truth for each simulated photon. In experiments, however, this information is lacking by definition. Nevertheless, there is a way to retrieve measures that should be complementary to the level of impurity in bursts from the autocorrelation of photon timestamps. In a nutshell, the autocorrelation function of all photon timestamps of bursts is calculated. Then, the resulting fluorescence autocorrelation function is fitted to a model inspired from fluorescence correlation spectroscopy of molecules freely diffusing in 3D in a confocal-based setup (see Equation 2 in Materials and Methods). One of the model free parameters, *<N>*, reports on the mean amount of molecules in the EDV at all times. The amount of molecules in the EDV at any given moment follows a Poisson distribution. Therefore, knowing the value of *<N>*, one may calculate the probability of having more than a single molecule in the EDV, *P (N > 1)*. Details of the herein procedure are given in *Materials in Methods*. Using this approach, we calculated *P (N > 1)* for all photons in a simulation, as well as *P_bursts_ (N > 1)* for photons in bursts, using Equations 3 and 4 (Materials and Methods), respectively. The best fit values of *<N>* and of *P (N > 1)* are reported in Appendix A.

Appendix A shows the calculated burst photon timestamp autocorrelation functions for the burst photons of the simulation with molecules in a concentration of 62 pM, freely diffusing in a numerical PSF model at a diffusion coefficient of 90 μm^2^/s. Appendix A also shows the best fitted results to the model of fluorescence autocorrelation of molecules freely diffusing in 3D in a confocal-based setup. As a first step, and as an additional validation, simulations using different diffusion coefficients, and a constant concentration, at a constant volume, should yield the same value of *<N>*, within error ranges. Appendix A shows that indeed the values of *<N>* retrieved from fitting, were the same in within the error ranges for different conditions in simulations having different diffusion coefficients, but the same concentration, per given burst analysis parameter values and per PSF model. *<N>* was consistently lower when using the numerical PSF model, compared to when using the Gaussian PSF model. That is expected, because the width of the numerical PSF model is smaller than the Gaussian PSF model, and hence less molecules cross it, in average (see Figure 1).

The probability of having more than a single molecule in the EDV, *P (N > 1)*, should, in principle, be somewhat analogous to the mean impurity level in bursts. Therefore, we decided to test whether the trends we observed in Figure 6 are similar if instead of assessing the mean impurity level in bursts we report the values of *P (N > 1)*. Appendix A reports on this assessment. The trends are similar, when considering the trends as a function of increasing the values of the burst search parameter, *F*. However, as for the trends when increasing the values other burst analysis parameters were opposite to the ones obtained against the mean impurity in bursts (Figure 6). Although these results may seem only partly reassuring, we believe that the retrieved values of *<N>* and *P (N > 1)* in this case are less credible, due to several technical reasons covered in details in the Discussion as well as in the Appendix A.

### 2.4. Quantification of Molecule Diffusion Times from Burst Photon Timestamp Autocorrelation and from Burst Width Analysis

An additional parameter that has not yet been assessed is the mean time of molecules traversing through a region of the EDV. Figure 2, Figure 3 and Figure 4 and Appendix A clearly show how burst analysis parameters (as well as experimental conditions such as the concentration and diffusion coefficient) influence the molecule positions from which photons are detected. The change of the molecular position dispersion hints on an additional expected influence on the duration of bursts, namely on burst widths or alternatively on the mean diffusion times, as retrieved from fits to the fluorescence autocorrelation functions. The expectation is that conditions that reduce the molecular position dispersion will also lead to a concomitant reduction in burst durations.

One possible value for comparison may be the mean diffusion time, as retrieved from the fits to the photon timestamp autocorrelation functions as in Appendix A. However, the values had quite large error ranges, which made all best fit values of the diffusion time useless due to their inaccuracy (see Appendix A; further details in Discussion). Another possible value for comparison relied on the average of all burst duration or widths, where a burst duration, or width, is defined as the time interval between the first and last photon timestamps identified as burst photons (see examples of the quantification of mean burst widths in Appendix A). The mean of all burst widths had different values for different burst analysis parameter values and different simulation conditions, with small error ranges. Appendix A, shows, for comparison, the values of the diffusion times compared with the values of the mean burst widths, to show the uselessness of the former, the usefulness of the latter, and their lack of correlation, both when using the Gaussian and numerical PSF models (Appendix A, left and right, respectively), although both time estimates are expected to be correlated. The values of these parameters are reported in Appendix A, as all of the quantities in this work are.

Therefore, we chose to show how the molecular position dispersion correlates with the mean burst widths. However, the expectation to observe a positive correlation between the two measures should be combined with another expectation that has nothing to do with the molecular position dispersion—increasing the values of burst analysis parameters that increase the amount of photons that are considered (e.g., the burst search parameter, *m*, the burst size threshold, *sz*, and to the burst width threshold, *w*), are expected to increase the mean burst widths, regardless of the molecular position dispersion. Additionally, regarding the instantaneous photon rate threshold *F*, increasing the photon rate in a burst, is expected to result in bursts with denser photons, and hence smaller burst widths. Overall, Appendix A shows that a positive correlation exists between the molecular position dispersion and the mean burst durations, when increasing the values of the instantaneous photon rate threshold *F*. Additionally, it can be seen in Appendix A that when changing the concentration, burst widths stay the same per given burst analysis parameter values, as expected. The values of burst widths increase with a decrease in the values of the diffusion coefficients, also as expected. Overall, the results shown in Appendix A may hint on a possible partial correlation of the molecular position dispersion with diffusion time, especially when it comes to modulating the values of the burst rate threshold, *F*.

### 2.5. Improving the Accuracy of Mean FRET Efficiency Estimation

We have shown that proper choice of burst analysis parameter values can greatly influence the occurrence of impure single-molecule bursts as well as their level of impurity. Next, we show how impure photon bursts may influence measurements based on ratios of burst photon counts.

In applications of confocal-based single-molecule fluorescence detection, histograms of ratiometric measures of photon counts are many times the main plots from which inferences are made. For instance, in single-molecule Förster resonance energy transfer (smFRET), FRET histograms are constructed out of FRET efficiencies of photon bursts, after the fluorescence signal is split onto two SPADs, using a dichroic mirror. The apparent FRET efficiency (also known as the proximity ratio) is calculated by taking the ratio of all photon detected in the acceptor fluorescence detection channel in the numerator and all photons detected both in the donor and acceptor fluorescence detection channels in the denominator. The shape of the histogram of the FRET efficiencies of all bursts can help make inferences on whether the sample included a FRET single population, or two (or more) distinct FRET sub-populations. This, in turn, is the power of smFRET—it helps define the amount of molecular or conformational species, as well as their mean FRET efficiencies.

#### 2.5.1. Simulation of smFRET Measurement of a Mixture of Two Species with Two FRET Efficiencies

Imagine that a sample contains a mixture of two types of molecules, yielding different FRET efficiencies, where the FRET efficiency of the first and the second sub-populations are 0.75 and 0.50, respectively. If a molecule from the second sub-population (the one with E = 0.50) traverses the EDV, it produces both donor and acceptor photons, with a FRET efficiency close to 0.50, with the deviation induced mostly due to lack of enough photons. This deviation, however, is not systematic. Now, imagine that while that molecule traverses the EDV, another molecule belonging to the first sub-population (the one with E = 0.75), also traverses a part of the EDV. There is a possibility that it will also emit a few photons, however since E = 0.75, there is a higher probability that these photons will be detected in the acceptor detection channel. Therefore, an E = 0.75 molecular impurity in an E = 0.50 burst, should systematically bias FRET efficiencies of the E = 0.50 population to values higher than 0.50.

In the previous sections, we have shown how different burst analysis parameter values influence the occurrence and level of burst impurity. More specifically, we have shown how increasing the value of the photon rate threshold, *F*, reduces both the occurrence of impure bursts and also reduces the level of impurity in the leftover impure bursts. Therefore, we can anticipate that fitting a FRET histogram of two sub-populations with a sum-of-two-Gaussians’ model will yield mean FRET efficiencies with values that may deviate from the ground-truth simulated values, and that the higher *F* will be, the smaller these differences will be. We therefore simulated the free 3D diffusion of fifteen molecules, at a concentration of 62 pM and with a diffusion coefficient of 90 μm^2^/s. We repeated this simulation four times, taking into account either numerical or Gaussian PSF models, and with simulations times of either 60 or 180 s. Then we allocated donor and acceptor photon timestamps for ten out of fifteen molecules, according to E = 0.75, and the leftover five molecules, according to E = 0.50. Then, we analyzed the results for bursts using a sliding window of *m* = 10 consecutive photons, with different values of the instantaneous photon rate threshold, *F* = 6, 11, and 21. We collected the FRET efficiencies of all bursts into FRET histograms and fitted them with a sum-of-two-Gaussians’ function. The results of this procedure are shown in Figure 8, for the 60 s simulation using the numerical PSF model, and the best fit values of all of these simulations are reported in Appendix A. Additionally, the fitting results are shown, after fixing the populations’ fraction to a fixed value of f = 0.6666, which is the value that was simulated (Appendix A).

Observe how the higher the value of the instantaneous photon rate threshold *F* was, the closer the best fit mean FRET efficiencies were to the ground-truth values. Clearly, using a large *F* value is a good practice when the accuracy of the retrieved values of the mean FRET efficiencies and the fraction of the FRET sub-populations is important. Although this assessment was well-controlled, we were aware that inaccuracy in retrieved mean FRET efficiencies can be caused by several other parameters (see Discussion). To strengthen our findings from this procedure, we implemented it this time on actual experimental smFRET results, in the next section.

#### 2.5.2. smFRET Experimental Results of a Mixture of Two Species with Two FRET Efficiencies

After showing how increasing the values of the burst rate threshold *F*, helps retrieving accurate mean FRET efficiencies, on well-controlled simulated data, next we show this feature also on well-controlled smFRET experimental results, measuring a mixture of two FRET DNA constructs with different mean FRET efficiencies. These results were taken from control smFRET experiments in a previous work of Lerner et al., where a mixture of two double-stranded DNA constructs with a sequence of the lacCONS promoter [16,17] and with the dyes Cy3B and ATTO 647N, as donor and acceptor dye pair, positions at different bases yielding different mean FRET efficiencies. Figure 9 summarizes a set of experimental results and their best fit results as well, for the measurements of one DNA construct having a mean FRET efficiency of ~0.68 (left), the measurements of a second DNA construct having a mean FRET efficiency of ~0.33 (center) and the best fit results of the experimental results of a mixture of the two DNA constructs, with a sum-of-two Gaussians function, and the deviation between the best fit mean FRET efficiency from these values (right) after burst search using the burst rate threshold F with values 6, 11, and 21.

Again, using elevated values of the photon rate threshold burst search parameter helps in retrieving accurate mean FRET efficiencies.

## 3. Discussion

The occurrence of impure bursts was theoretically suggested to occur in the seminal work of Eggeling et al. describing the theory of burst size distributions (such bursts were referred to as *multi-molecule events*) [8]. However, to the best of our knowledge, a systematic assessment of burst impurity has not yet been performed. In this work, we have systematically tested the effect of different burst analysis parameters on the underlying diffusing molecules. Specifically, we focused on two main results: 1) How close were the molecules, that were detected as burst photons, to the center of the effective detection volume, we referred to as the molecular position dispersion, and 2) how often and how well were photon bursts originating from single molecules. We have shown that increasing the values of burst analysis parameters helps in reducing the molecular position dispersion. However, the parameters, that when increasing their values, mostly influence the reduction of the molecular position dispersion were the photon rate threshold, *F*, used in the burst search procedure and the burst size threshold used afterwards in burst selection. Then we have shown that increasing the value of *F* helps in reducing the amount of impure bursts and their level of impurity. Increasing the value of the burst size threshold, on the other hand, introduces an increase in burst impurity. Additionally, we have shown that when it comes to increasing the values of the photon rate threshold, burst impurity was positively correlated with the molecular position dispersion, while on the other hand, for increasing the values of burst analysis parameters that lead to larger burst sizes (the burst search parameter *m*, or the burst size threshold), burst impurity correlated negatively with the molecular position dispersion.

Then, we have shown that measuring the molecules at lower concentrations, help reduce both the molecular position dispersion and burst impurity. However, sometimes, it might not be practical. The lower the concentration is, the longer data acquisition will take until a high enough number of selected bursts is achieved. When performing smFRET experiments, if the labeled molecules are at a concentration of 50–100 pM, proper data acquisition of enough legible bursts (bursts that passed the burst search and selection procedure) can take 5–10 min. However, decreasing the concentration by an order of magnitude will increase acquisition time by an order of magnitude, which for some applications, and for some experimentalists, may be considered too long for a single measurement. State-of-the-art multi-spot single-molecule spectroscopy allows to mitigate this problem, by parallelizing a large number of independent SMFD measurements [11].

We have also assessed what would be the effect of decreasing diffusion coefficient values on the resulting single-molecule bursts. Decreasing values of the diffusion coefficient had a clear effect both on the reduction of the molecular position dispersion and on diminishing burst impurity, however their correlation is less clear.

Certain concerns may be raised as to how the simulations and their quantifications were performed. Therefore, we tested these concerns one by one and reported our results regarding them.

In this work we heavily relied on the comparisons of the results of burst analyses and the underlying molecules with their positions when they emitted photons that were identified and selected by the burst analysis procedure. We used this knowledge to calculate the exact burst impurity levels as well as quantities characterizing all bursts, such as the occurrence of impure bursts, their mean level of impurity as well as the molecular position dispersion. The results show the dependence of the different burst analysis parameters on both the regions in space from which photons were emitted, detected, and selected by the burst analysis procedure, as well the possibility of observing burst photons originating not solely from a single molecule. These parameters influence two parameters that are highly useful in fluorescence correlation spectroscopy (FCS)—the diffusion time through the EDV, *τ_D_*, and the mean number of molecules in the EDV, at any given moment, *<N>*. However, while FCS is useful when measuring higher concentrations (nM and above), in which the noisy signal has a clear mean, and the information is found in the temporal fluctuations about the mean signal, in concentrations used in confocal-based SMFD (<100 pM) there is not one mean signal, but rather two characteristic Poisson processes, with two different mean rates (the BG process, when no molecule crosses the EDV, and the signal process, when a molecule crosses it). Accordingly, analysis of fluorescence autocorrelation curves of photon timestamps only from bursts was a more promising route, however, their analysis required assuming the EDV has a constant Gaussian shape. The EDVs, following burst analysis, as are shown from molecular position histograms in this work, change shape and width. Therefore, estimates retrieved from best fit values of *<N>*, such as *P (N > 1)*, the probability of having more than a single molecule in the EDV (see Equations (3) and (4)), were biased and less credible as estimates of burst impurity derived directly from knowing the molecular identity and position of each photon timestamp. Indeed, the trends we reported as a function of the actual burst impurities were sometimes different than the ones using *P (N > 1)*. In addition, the best fit values of the diffusion time had large error ranges, rendering them less useful when trying to quantify relations to the diffusion time through the EDV. Burst widths, however, served as an accurate quantity for that purpose.

Another concern might be raised from the use of a numerically calculated PSF model, deviating from the Gaussian approximation that is heavily used when describing confocal-based measurements such as FCS or confocal-based SMFD. Therefore, we performed all of our simulations using either Gaussian or numerical PSF models, constructed on the basis of them having similar energies, rather than similar widths. While for most assessments, we report similar trends when using either of the two PSF models, their absolute values are different, as the Gaussian PSF model was wider than the numerical PSF model.

We ran most of the simulations for a duration of 60 s. A concern might be raised that 60 s of burst simulations might not include enough bursts or burst photons for attaining statistically meaningful estimation of quantities. Therefore, we also performed simulations lasting 180 s, and then compared the values of all the retrieved and calculated quantities, and compared them to see whether the values are the same in 60 and 180 s simulations, within error ranges. Appendix A reports on this comparison. It clearly shows how all the retrieved quantities (mean fraction of impure burst photons, fraction of impure bursts, the molecular position dispersion in z and x coordinates, the FCS-derived parameters *τ_D_* and *<N>,* and the mean burst width) are the same within the error ranges, for all burst analysis parameter values tested in this work. It means that the values acquired in the 60 s simulations are representative, and that longer simulation times may increase the accuracy of the retrieved values. In that context, it is worth mentioning that further increasing the simulation time was impossible due to the large file sizes produced for the molecules’ 3D trajectories (~50 Gb per 60 s, using the simulation conditions described in Materials and Methods).

As a proof of concept, we have shown the effect of impure photon bursts on the accuracy of the retrieved mean FRET efficiency values, from histograms of burst-wise FRET efficiencies. In situations when a mixture of molecular species is under measurement, bursts of one molecular species that are contaminated by photons arising from molecules of the other species can influence the overall mean FRET efficiency retrieved from fits to the FRET histogram. The recommendation to use elevated values of the instantaneous photon rate threshold, *F*, proved beneficial in increasing the accuracy of the retrieved mean FRET efficiency (Figure 8, Figure 9 and Appendix A).

In this context, it is important to mention that burst impurity may be only one reason influencing the accuracy of the retrieved mean FRET efficiencies in the analysis of confocal-based smFRET measurements. Although we have shown that increasing the values of the burst search parameter *m*, or the values of the burst search thresholds, lead to an increase in burst impurity, we did not test the accuracy of the mean FRET efficiencies retrieved from Gaussian fitting. That is because increasing the values of these burst analysis parameters increases the sizes of bursts, but reduces the overall amount of bursts drastically. Reduction of the amount of bursts in FRET efficiency histograms makes the comparison of the performance of Gaussian fitting hard, because the comparison requires keeping the amount of items (bursts in our work) in compared histograms, relatively similar. Changing the values of the photon rate threshold *F*, produces histograms with slightly different numbers of bursts. Additionally, it is worth noting that a Gaussian model describes a FRET efficiency population only approximately, mainly due to the fact that FRET efficiency populations tend to be skewed towards a value of 0.5, while a Gaussian model is totally symmetric about its mean value.

Retrieving accurate mean FRET efficiencies is highly important in the application of smFRET to retrieve information on accurate donor-acceptor distances, in its use as spatial restraints for integrative structural modeling (reviewed by Lerner and Cordes et al. [1]). In this context, it is worth noting that impure photon bursts may also bias other estimates based on photon counts in a burst, and their ratios, such as, for instance, in single-molecule fluorescence anisotropy. In fluorescence anisotropy, much like in smFRET, the fluorescence signal is split onto two SPADs, only by a polarizing beam splitter, rather than by a dichroic mirror. The fluorescence anisotropy is a ratio of photon counts, and hence is influenced by single-molecule impurity from other molecules with a different mean fluorescence anisotropy. Therefore, the recommendation to use elevated values of *F* in burst analysis of SMFD measurements is of general use when analyzing single-molecule fluorescence bursts of freely diffusing molecules in SMFD measurements.

Our survey did not cover all characterizations of single molecule bursts in general (e.g., different molecular brightness, BG rates, burst rates, bursts from recurring molecules) and parameters of burst-wise smFRET in particular (e.g., estimating the fraction of FRET efficiency populations and the effect of FRET populations with different molecular brightness, bursts of molecules dynamically interconverting between different FRET values). Additionally, we did not assess the effects of burst search parameters used in special experimental modalities, such as in alternating laser excitation (ALEX) [9,18] experiments (e.g., dual-channel burst search [10], or ALEX-2CDE filters [19]). That is mainly because we wanted to assess the most general features of SMFD confocal-based. Nevertheless, the abovementioned parameters can be the subject of future systematic assessments. In that context, it is worth mentioning the relatively approach for the analysis of rapid FRET dynamics out of photon-by-photon hidden Markov modeling [20,21]. It would certainly be interesting to test the effects of impurity on proper estimation of FRET dynamics.

While we focused on molecules freely diffusing in 3D, the next logical steps would be to test the performance of the burst search analysis on 2D diffusion (for membrane proteins), 1D diffusion (for filament-associating proteins), and in situations where not only diffusion occurs but also convection and perhaps flow. These features and others will be the subject of further investigation in future work.

In summary, our systematic assessment helped reduce the arbitrariness of parameter value choice in the analysis of bursts from confocal-based SMFD experiments. Impure bursts exist. Impurity levels depend on the molecular position dispersion, and hence on the part of the EDV out of which photons are selected by the burst analysis procedure. A treatment as simple as elevating the value of the photon rate threshold, *F*, can assist in diminishing the extent of impurity and its possible effects on accurate quantitative estimations performed on SMFD experimental results.

## 4. Materials and Methods

We performed all 3D diffusion simulations using PyBroMo (Python Brownian motion) simulations [13] (https://github.com/OpenSMFS/PyBroMo/releases/tag/0.8.1; was utilized in previous works [21,22,23,24]). In the simulation of diffusion trajectories, we recorded the *x*, *y,* and *z* positions of each diffusing molecule at each moment and advanced the molecular positions in intervals of 200 ns. We tested 15 molecules in different concentrations (different volumes of the rectangular box), different diffusion coefficients, with simulations that lasted either 60 or 180 s, and against two point-spread-function (PSF) models: Numerically-calculated PSF. It was derived from a model of a realistic PSF of a typical 60x water immersion objective with a numerical aperture of 1.2, with a sample mounted on top a 150 μm coverglass and with sample excitation at a wavelength of 532 nm (see Figure 1). We modeled the excitation PSF using a vectorial electromagnetic simulation, PSFLab [15]. This model includes effects of refractive index mismatch as well as mismatch between objective lens correction and coverglass thickness. The model we used is based on the excitation PSF taken to the power of two, to calculate the multiplication of the excitation PSF with the detection PSF [25,26], that is assumed to be similar to the excitation PSF. In general, the multiplication of the excitation and detection PSFs should be convolved with the pinhole profile [25,26]. The convolution with the pinhole function was not considered as the simulations mimic measurements with overfilling of the objective lens back aperture, rather than underfilling. A convolution with the pinhole profile would be required in cases of underfilling the back aperture of the objective lens.A Gaussian-shaped PSF with standard deviations of 180 nm in the *x* and *y* directions and 880 nm in the z direction.

The PSF models were calculated to have comparable energies, rather than widths. Therefore, the full width at half maximum (FWHM) of the numerical PSF was smaller than its Gaussian counterpart. Appendix A summarizes the different simulation conditions we tested.

In the next step, the instantaneous emission rates of each molecule at each moment in the Brownian motion simulation were calculated by evaluating the intensity of the PSF at different points in space corresponding to the molecular positions in the diffusion trajectories. After the diffusion simulation and the calculation of the instantaneous emission rates, photons were generated from a Poisson process using the instantaneous emission rates, assuming the photon rate at the maximum intensity of the PSF (the molecular brightness) was 200 KHz. Each detected photon was assigned the molecule from which it originated. Afterwards, Poisson-distributed BG timestamps were also added. The simulated mean BG rate was 2.3 KHz. In the end of the simulation, a photon HDF5 [22] file was constructed, including all photon timestamp assignments, as if it was a file containing experimental data. Therefore, next, each simulated photon HDF5 file was analyzed for identifying single molecule photon bursts, using the FRETbursts single-molecule photon detection analysis suite [14].

The burst analysis parameters that were tested and their values were: i) The number of consecutive photons, *m*, in the burst search sliding window (values 5, 10, 15, and 20, at a constant photon rate threshold, *F* = 6), ii) the photon rate threshold, *F*, defined as the minimal multiplier of the BG rate that can be considered a signal (values 3, 6, 11, 16, and 21, at a constant *m* = 10 consecutive photons), iii) minimal burst size thresholds (values 10, 20, 40, and 80, for constant *m* = 10 and *F* = 6; burst size threshold value of 10 is already included for *m* = 10 and *F* = 6, since by definition the identified bursts include at least *m* = 10 consecutive photons), and iv) minimal burst width thresholds (values 0.0, 0.5, and 1.0 ms, for constant *m* = 10 and *F* = 6; burst width threshold value of 0.0 ms is already included for *m* = 10 and *F* = 6, since by definition the identified bursts include all bursts, with all burst widths).

Simulations of single-molecule Förster resonance energy transfer (smFRET) measurements of a mixture of two FRET sub-populations were also carried out, with the following parameter values: fifteen diffusing molecules were simulated (at a concentration of 62 pM, with a diffusion coefficient of 90 μm^2^/s), with ten of them having a mean FRET efficiency of 0.75, and five with a mean FRET efficiency of 0.5. The two groups of molecules were simulated in the same rectangular box and had the same diffusion coefficient and the same molecular brightness (photon rate of 200 KHz at the maximum intensity of the PSF). The simulated BG rates of the donor and acceptor fluorescence detection channels were 1.5 and 0.8 KHz, respectively. Additionally, assessing the accuracy of the retrieves mean FRET efficiency values from smFRET of a mixture of two subpopulations was also carried out on experimental data, acquired in previous works [21]. These were microsecond alternating laser excitation (μsALEX) smFRET measurements of two FRET constructs with Cy3B and ATTO 647N as the donor and acceptor dyes, respectively, labeling a pair of bases in a dsDNA with a sequence of the lacCONS promoter. The FRET histograms of bursts from these measurements were achieved on all bursts that had at least 20 acceptor photons, after directly exciting it with a 635 nm laser. The photons arising after donor excitation were analyzed as the simulation results were analyzed.

In addition to FRETbursts, fluorescence autocorrelation functions of burst photons timestamps were performed using the Python Pycorrelate analysis suite [27]. All nonlinear regression fits of FRET histograms and of fluorescence autocorrelation functions were performed using the non-linear least-squares algorithm in the Python suite, lmfit [28]. The model used for fitting simulated histograms of FRET efficiencies were a sum-of-two-Gaussians. The models used for fitting experimental histograms of FRET efficiencies were a sum-of-two-gaussians, in the case of two-subpopulations of FRET efficiencies, and a single Gaussian, in the case of a single population of FRET efficiencies. The model used for fitting fluorescence autocorrelation functions was a model of fluorescence autocorrelation of molecules diffusing in 3D in a Gaussian-shaped PSF. The model is as in Equation 2.
*G(τ) = {<N>[1+(τ/τ_D_)]^2^[1+(1/κ^2^)(τ/τ_D_)]}^−1^*(2)
where τ is the lag time between photons, in seconds, *G(τ)* is the autocorrelation function, *<N>* is the average number of molecules traversing the EDV at any given moment, *τ_D_* is the mean diffusion time through the EDV, and *κ* is the z-to-x ratio of the EDV (taken with a value of 7, in the fits of the autocorrelation functions to this model). The best fit *<N>* was also used for the calculation of the probability for having more than one molecule crossing the EDV, according to Equation 3 in the case of autocorrelation of all photons, and according to Equation 4 in the case of autocorrelation of only the photons inside bursts.
*P(N>1) = 1 – Pois(0) – Pois(1)*(3)
*P_Burst_(N>1) = [1 – Pois(0) – Pois(1)]/[1 – Pois(0)]*(4)
where *Pois (0)* and *Pois (1)* are the Poisson probabilities of having no molecules (0), or one molecule (1) cross the EDV, respectively. The calculation for photons inside bursts is different, since by definition, a burst of photons originated from at least one molecule, which drops the probability term of having no molecules in a burst. Equation 3 was used in this work for calculating the probability of >1 molecule for all photons in a simulation, while Equation 4 was used when only burst photons were taken into account.

In addition to these parameters, the following parameters were calculated out of the analyses of the simulations:The fraction (percentage) of photonic impurity—all photons of bursts not originating from the main burst molecules, divided by all the burst photons in the simulation.The fraction of impure bursts—the fraction of bursts that included photons from more than a single molecule.The mean burst impurity—the mean over all bursts of the fraction of burst photons not originating from the main molecule in the burst.The molecular position dispersion in x, y, and z—standard deviation of all molecular positions in each burst photon, calculated for each dimension.The mean burst width—the mean of all burst widths/durations, where the burst duration is calculated as the time interval between the first and last photon timestamps in a burst.

The values of the abovementioned calculated and best fit parameters were used in figures overall this work, and are summarized in Appendix A. In that context, it is important to mention that the quantities reported in Appendix A and in this work, from all simulation conditions shown in Appendix A, rely on calculations that were performed in Jupyter notebooks, for which we also produced raw figures. In this work we show only a representative portion of these figures (Figure 2, Figure 7, Figure 8, Figure 9, Appendix A), for the sake of presenting the work. We deposited all the raw figures of showing all quantities of all simulation conditions in Zenodo [29].

All of the PyBroMo and FRETbursts [14] code used here were documented in Jupyter notebooks, that were deposited in Zenodo [30]. These simulations produced photon HDF5 files, that hold all the simulated molecule positions, photon timestamps and photon identity (for smFRET simulations—either donor or acceptor detection channels), in files with names beginning with ‘pybromo_’, ‘times_’ and ‘smFRET_’, respectively. The photon HDF5 files of the photon timestamps and identities were deposited in Zenodo [31]). The molecule diffusion trajectory photon HDF5 files were not deposited, due their large size. These files can be reproduced by using the ‘*PyBroMo - 1-. Simulate 3D trajectories - single core - different 3D diffusion simulation conditions.ipynb*’ Jupyter Notebook (deposited in Zenodo [30]), where we documented the input parameter values for all simulation conditions performed in this work.

All photon HDF5 files carrying photon timestamps were analyzed by FRETbursts [14], by first estimating the BG rate, and then, by using the sliding window algorithm [6,11,12]. Then, photons from single-molecules were identifying as having instantaneous photon rates larger than *F* times the BG rate. Identified bursts, were further selected by using different types of burst selection criteria. Then, per each set of burst analysis results, we tested which was the molecule that produced each burst, its level of impurity (how many photons in the burst originated from molecules other than the main one), and the positions of the molecule when it emitted the burst’s photons. All the details are specified in the text above as well as in the Jupyter notebooks deposited in Zenodo [30].

All figures in this work were produced either by matplotlib [32] in the Jupyter notebooks, or by OriginLab Origin 2018.

## Figures and Tables

**Figure 1 molecules-24-02557-f001:**
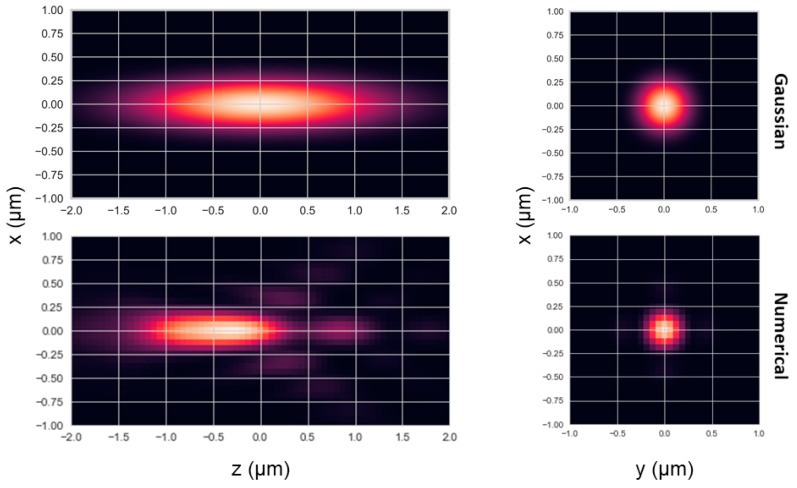
Models of the point-spread function (PSF). The PSF was modeled either as Gaussian-shaped or numerically calculated by using the model of an excitation PSF calculated with PSFLab [15] for a typical 60x water immersion objective with a numerical aperture of 1.2, with a sample mounted on top a 150 μm coverglass and with sample excitation at a wavelength of 532 nm, where the PSF that was used was the calculated excitation PSF, taken to the power of two. For more information, see Materials and Methods.

**Figure 2 molecules-24-02557-f002:**
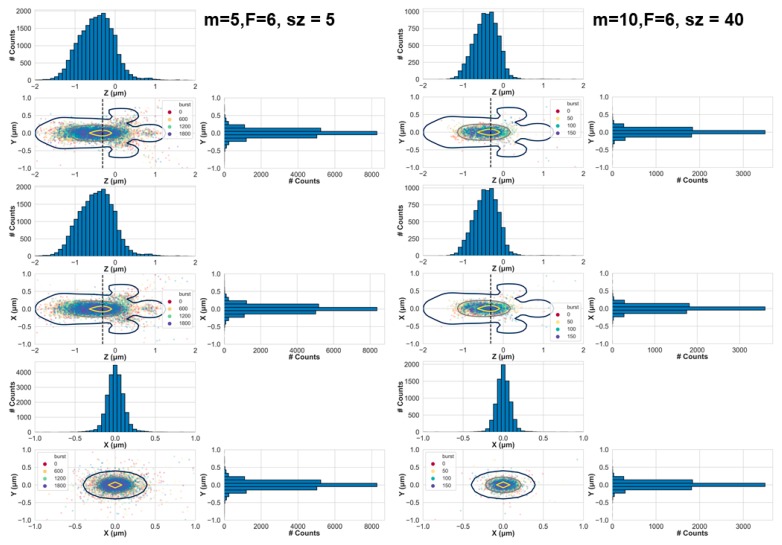
The positions of diffusing molecules when they emitted photons that were detected and selected by the burst analysis, either with minimal burst analysis parameter values (*m* = 5 and *F* = 6; left panels) or with stringent burst analysis parameter values (*m* = 10, *F* = 6 and burst size threshold, *sz* = 40; right panels). In the top, central, and bottom panels we show the 2D projections at the yz, xz, and xy planes, respectively. Each dot in the scatter plots is an emitted photon. These results are for the simulation of molecules in a concentration of 62 pM, where the diffusion coefficient of the molecules was 90 μm^2^/s. The colors of the points correspond to the burst number out of the overall number of bursts. In each panel, the 1D projections are also shown as histograms. The black, brown, and yellow contour lines align the position of the numerical PSF model (see shapes of PSF models in Figure 1).

**Figure 3 molecules-24-02557-f003:**
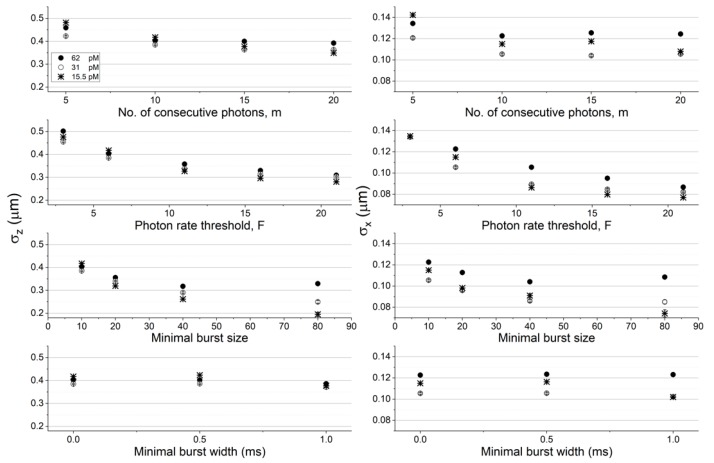
The molecular position dispersion as a function of burst search criteria and experimental conditions (different concentrations). Shown are the standard deviation of molecular positions in the z (left) and x (right) coordinates (the values in the y coordinate are the same as the ones in the x coordinate, in within the error ranges), when they emitted photons that were detected and selected by the burst analysis. The error values were calculated as the uncertainty of the standard deviation. All values are reported in Appendix A. The assessment of the molecular position dispersion here is shown as a function of different concentrations for molecules diffusing with a constant diffusion coefficient of 90 μm^2^/s, and in simulations using the numerical PSF model.

**Figure 4 molecules-24-02557-f004:**
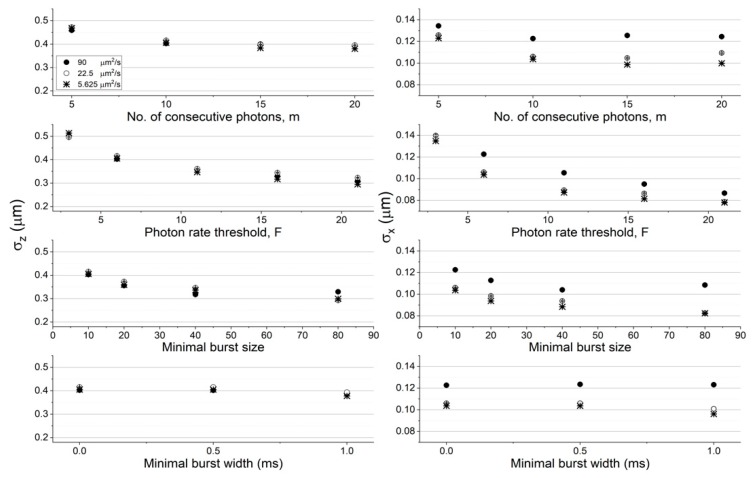
The molecular position dispersion as a function of burst search criteria and experimental conditions (different diffusion coefficients). Shown are the standard deviation of molecular positions in the z (left) and x (right) coordinates (the values in the y coordinate are the same as the ones in the x coordinate, in within the error ranges), when they emitted photons that were detected and selected by the burst analysis. The error values were calculated as the uncertainty of the standard deviation. All values are reported in Appendix A. The assessment of the molecular position dispersion here is shown as a function of molecules diffusing with different diffusion coefficients, at a constant concentration (62 pM), and in simulations using the numerical PSF model.

**Figure 5 molecules-24-02557-f005:**
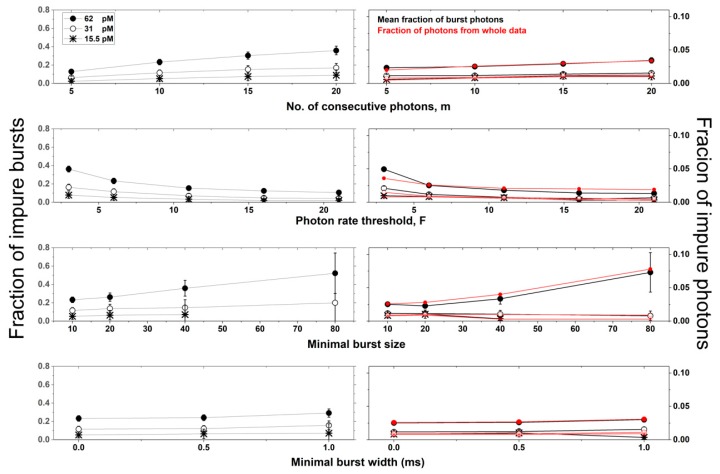
The occurrence and level of impure bursts as a function of burst search criteria and concentrations. Different burst analysis parameter values for different concentrations of molecules. The relative occurrence of impure bursts (left) was calculated as the fraction of bursts with an impurity level larger than 0 (error ranges calculated as the 95% confidence intervals), as the fraction of non-single-molecule bursts, and hence as the fraction of impure bursts. The level of impurity (right) was calculated as either the mean of all burst impurity levels (black; error ranges calculated as the standard error) or as the fraction of impure photons from all bursts relative to all burst photons (red; no error ranges, as the calculation was performed over all photons). The assessment is shown as a function of different concentrations for molecules diffusing with a constant diffusion coefficient of 90 μm^2^/s, and in simulations using the numerical PSF model.

**Figure 6 molecules-24-02557-f006:**
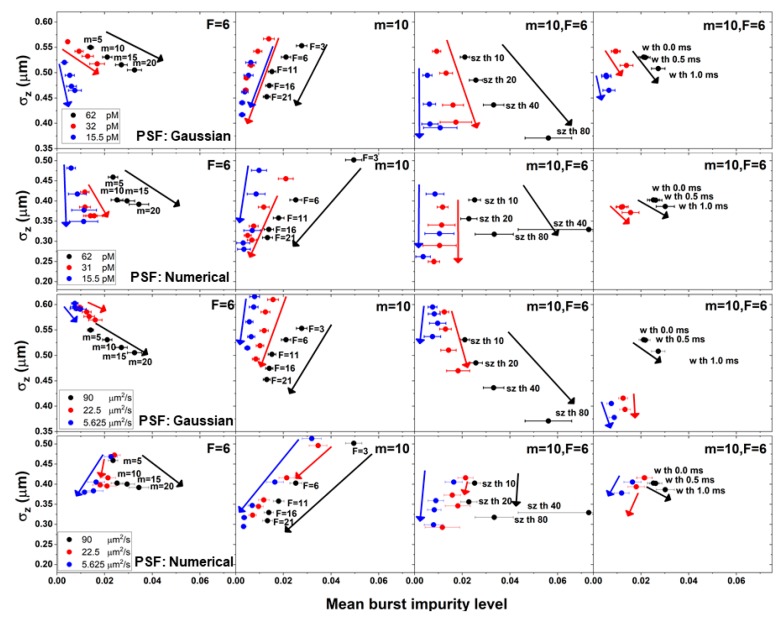
The correlation of burst impurity with molecular position dispersion. Different burst analysis parameter values for different concentrations of molecules. The mean burst impurity levels (error ranges calculated as the standard error) were compared against the molecular position dispersion in the z coordinate (error ranges calculated as the uncertainty of the standard deviation), as a function of different burst analysis parameter values (from left to right: varying *m* values, varying *F* values, varying burst size threshold values, and varying burst width threshold values), for different simulation conditions (from top to bottom: Different concentrations at a constant diffusion coefficient value of 90 μm^2^/s in simulations using the Gaussian PSF model, different concentrations at a constant diffusion coefficient value of 90 μm^2^/s in simulations using the numerical PSF model, different diffusion coefficients at a constant concentration of 62 pM in simulations using the Gaussian PSF model, and, different diffusion coefficients at a constant concentration of 62 pM in simulations using the numerical PSF model).

**Figure 7 molecules-24-02557-f007:**
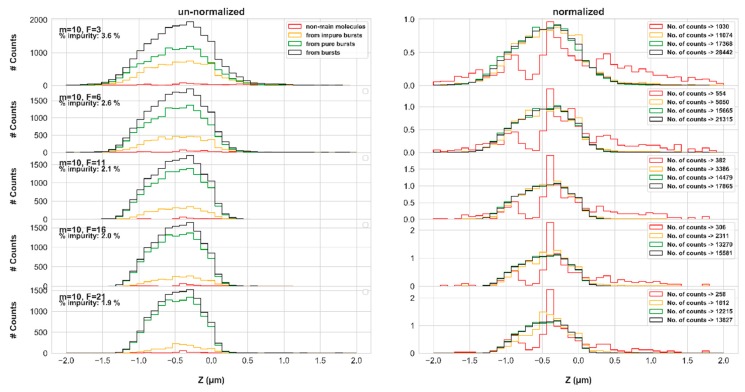
The molecular positions of pure and impure bursts photons, as a function of varying instantaneous photon rate threshold values, *F* (numerical PSF model)—shape and amplitude. We the histograms of molecular positions in the *z* coordinate of impure photons (red), burst photons of impure bursts (yellow), of pure bursts (green), and of all bursts (black), both un-normalized (left) to assess the weight of burst impurity, and normalized (right) to assess the histogram shapes. These results refer to the simulations in concentration of 62 pM and diffusion coefficient of 90 μm^2^/s, using the numerical PSF model.

**Figure 8 molecules-24-02557-f008:**
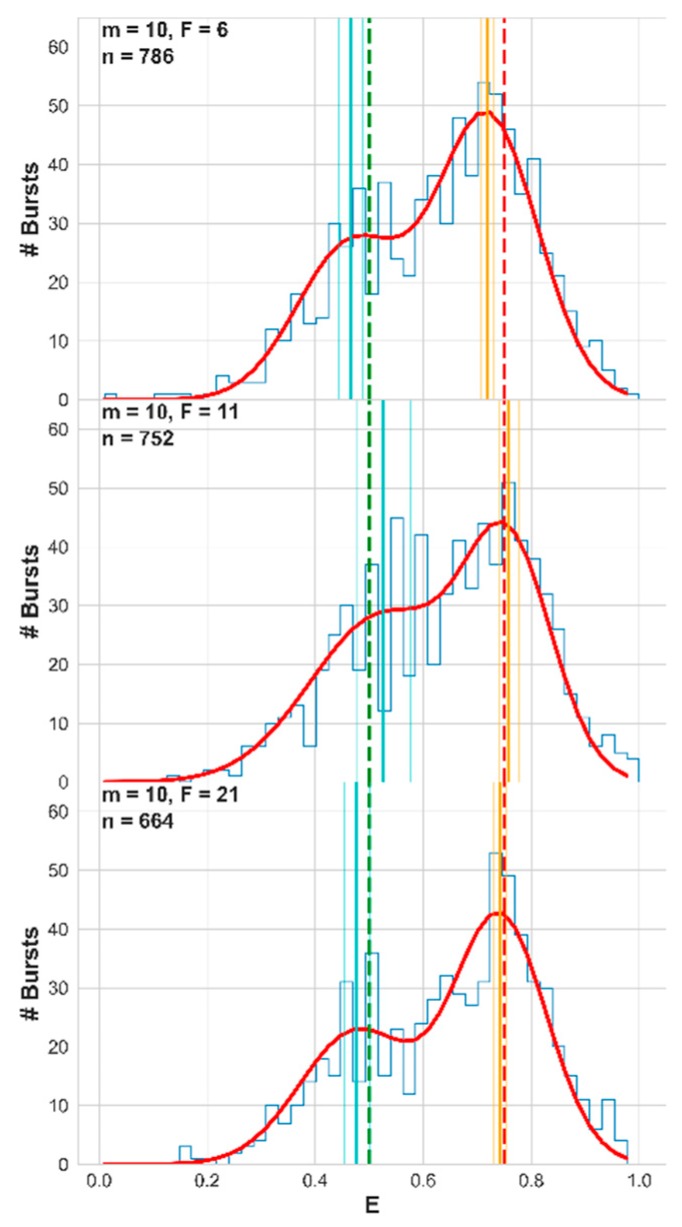
Simulations of single-molecule Förster resonance energy transfer (smFRET) experiments with two FRET efficiency subpopulations—increasing the value of the photon rate threshold, *F*, improves the accuracy of the retrieved mean FRET efficiency. From top to bottom, each panel shows the resulting FRET histogram (blue), the best fit sum-of-two-Gaussians (red), the best-fit mean FRET efficiencies (orange and cyan vertical lines; dimmer lines show the error ranges), and the simulation ground-truth mean FRET efficiency values (dashed red and green vertical lines). These results are for the 60 s simulation of molecules a concentration of 62 pM, where the diffusion coefficient of the molecules was 90 μm^2^/s, using the numerical PSF model, and the molecules were split to 10 with E = 0.75 and 5 with E = 0.5. The number of bursts in each histogram is also reported in each panel. The best fit values and the fitting error values are also reported in Appendix A.

**Figure 9 molecules-24-02557-f009:**
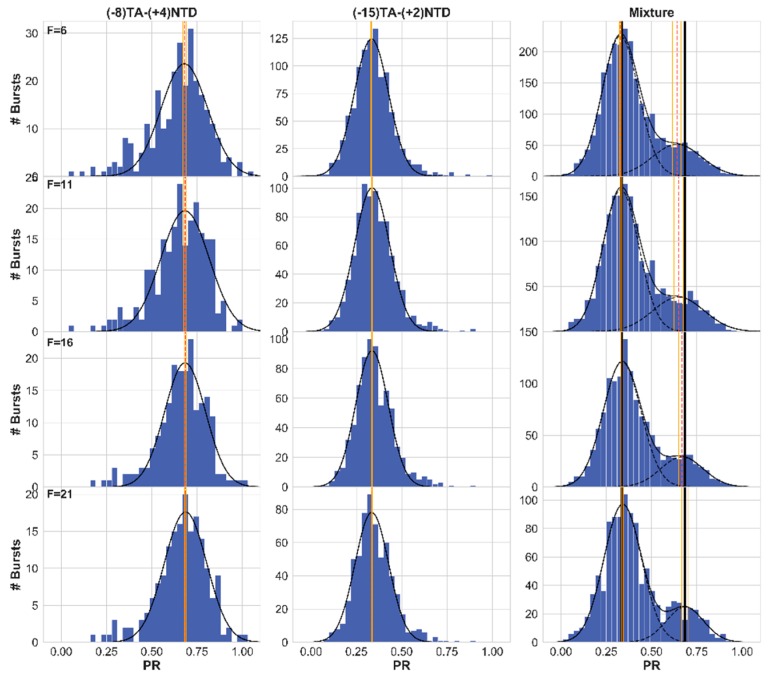
smFRET measurements of a mixture of two FRET constructs with different mean FRET efficiencies—increasing the value of the photon rate threshold, *F*, improves the accuracy of the retrieved mean FRET efficiency. From top to bottom, each panel shows the resulting FRET histogram (blue), the results of the best fit models (black), the best-fit mean FRET efficiencies (dashed red vertical lines; orange vertical lines show the error ranges), and the expected mean FRET efficiencies of each FRET sub-population (black vertical lines; calculated from the best fit results of fits of single FRET population to single Gaussian functions in the left and center panels). These results are for the measurements of FRET lacCONS promoter construct labeled with Cy3B at the nontemplate strand and ATTO 647N at the template strand at registers +4 and -8, respectively (left), +2 and -15, respectively (center) and a measurement of their mixture (right). The best fit values are also reported in Appendix A.

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
