# Peer review of "Systematic Assessment of Burst Impurity in Confocal-Based Single-Molecule Fluorescence Detection Using Brownian Motion Simulations"

_molecules, 2019, doi:10.3390/molecules24142557_

Reviewer 1 Report

In the presented manuscript Hagai and Lerner present a detailed, quantitative assessment of the burst analysis parameters for single-molecule detection. Single-molecule FRET is a widespread, popular technique, and photon burst analysis is a powerful tool of signal analysis for single-molecule detection. As such, rational basis for selection of the analysis parameters is of potential interest. The presented work has the potential to become a useful guide for experimentalists in the field of single-molecule spectroscopy who wish to perform burst analysis and have quantitative grounds for selection of their threshold parameters. As such, the work could be well-suited for publication in Molecules.

However, currently the format of the manuscript is reminiscent rather of a master thesis than a journal paper. This applies both to the paper organization and the type of presented data. In my opinion the following revision steps are necessary to make the paper of potential use:

1) The data should be presented in a (much) more concise way. Rather than presenting all intermediate results, fewer key figures summarizing the outcome would be far more useful.

2) In the text the outcomes/recommendations should be summarized at one particular place (towards the end), while presentation of the particular results should be accompanied with discussion rather than conclusions. (Example: page 15, line 445 and further on, ...the recommendation that this work provides... would be better placed in a concluding section at the end).

Finally, what would truly benefit the work would be application to/discussion of real experimental data. To this purpose data from the group previous work (Ref. 18, Ref. 11, and other) could be very well used. As in real experiment other, less-controlled factors than presented in Table 1 are of importance, it would be very interesting to discuss the burst analysis parameter influence on measured data analysis. This would make the work way more convincing to potential readers from the single-molecule field.

Detailed comments:
In several figures, suddenly non-monotoneous behaivor occurs. Take for example the panel in 3rd row, 2nd column (non-SM bursts vs minimal burst size fraction). At the 15pM conc. at burst size 20 the non-SM burst fraction is the largest, which seems peculiar. Do the authors hav physical explanation for such non-monotonous behavior? Or are such points results of unconverged simulations? The latter case would be very worrying!

What is the origin of the peculiar shape of the 90% contour in Figures 1, 2 and 11? In particular, what is the origin of the two lobes in the X/Y vs Z projection? Shouldn't the shape be related to the 3D point-spread function of the illumination?

The labels and legends in the figures should be larger, to be more easily legible (at least such as in Fig. 3). It would help if all the quantitites (such as the 90% contour) were described in the caption instead in the text only.

page 17, line 526: sentence 'However, the parameters that when....' seems impossible to understand.

page 17, line 540: multisport SMS

page 17, line 546 and further: motional narrowing has a fixed, different meaning in traditional spectroscopy, recommend rephrasing

Author Response

Reviewer No. 1

In the presented manuscript Hagai and Lerner present a detailed, quantitative assessment of the burst analysis parameters for single-molecule detection. Single-molecule FRET is a widespread, popular technique, and photon burst analysis is a powerful tool of signal analysis for single-molecule detection. As such, rational basis for selection of the analysis parameters is of potential interest. The presented work has the potential to become a useful guide for experimentalists in the field of single-molecule spectroscopy who wish to perform burst analysis and have quantitative grounds for selection of their threshold parameters. As such, the work could be well-suited for publication in Molecules.

We wish to thank reviewer No. 1 for his positive feedback and review.

 However, currently the format of the manuscript is reminiscent rather of a master thesis than a journal paper. This applies both to the paper organization and the type of presented data.

In that context, we wish to ask reviewer No. 1 to understand that while we did our best to present the content of this paper in a more concise fashion, other reviewers asked us to add additional details that were missing in the previous version of the manuscript. Something that makes the manuscript comprehensive and hence long again. We did do our best to shorten the content, to minimize the amount of figures in the main text, to unify summarizing figures to single ones, as well as to reformat the flow of the content in this work. Additionally, we changed the paper’s title so it will better fit with the findings of this work.

 In my opinion the following revision steps are necessary to make the paper of potential use:

1) The data should be presented in a (much) more concise way. Rather than presenting all intermediate results, fewer key figures summarizing the outcome would be far more useful.

We would like to thank reviewer No. 1 for this comment. As one can notice, now the figures in the main text include only the most important raw results (and only of a few out of all the simulations we now ran) and summarizing paneled figures. As we mentioned above, we were asked to perform longer simulations, simulations also with a Gaussian PSF model, we were asked to test parameters derived from FCS and from burst width analyses, as well as other requests. All of the above added up additional content that had to be presented and as a result was included in the flow of the main text. Additionally, many representative figures were diverted into the Supplementary Materials. To allow readers of the work access to all raw figures and raw data, we added all raw figures produced by the Jupyter notebooks, on top of the existing repositories of the raw data and its analyses. As part of shortening the content, we moved many of the intermediate explanations we previously had in the Results chapter, to the Discussion chapter. We ask Reviewer No. 1 to judge the current format of the manuscript not only following his review report, but also the review report of the other reviewers.

2) In the text the outcomes/recommendations should be summarized at one particular place (towards the end), while presentation of the particular results should be accompanied with discussion rather than conclusions. (Example: page 15, line 445 and further on, ...the recommendation that this work provides... would be better placed in a concluding section at the end).

We followed the reviewer’s recommendations and moved almost all intermediate discussions and conclusions that appeared in the Results chapter, to the Discussion chapter.

Finally, what would truly benefit the work would be application to/discussion of real experimental data. To this purpose data from the group previous work (Ref. 18, Ref. 11, and other) could be very well used. As in real experiment other, less-controlled factors than presented in Table 1 are of importance, it would be very interesting to discuss the burst analysis parameter influence on measured data analysis. This would make the work way more convincing to potential readers from the single-molecule field.

The discussion chapter now includes all of the above, and much more. Additionally, on top of the proof of concept analysis of simulated single-molecule FRET data, we added also analysis of actual experimental single-molecule FRET data, we acquired in our previous work.

Detailed comments:

In several figures, suddenly non-monotoneous behaivor occurs. Take for example the panel in 3rd row, 2nd column (non-SM bursts vs minimal burst size fraction). At the 15pM conc. at burst size 20 the non-SM burst fraction is the largest, which seems peculiar. Do the authors hav physical explanation for such non-monotonous behavior? Or are such points results of unconverged simulations? The latter case would be very worrying!

Now, the Results chapter does refer to these deviations from monotonicity, in the text. As a sanity check, we checked our code (again) and corrected a few minor bugs. Additionally, we have now added error ranges, which can help in deciding whether non-monotonic trends were statistically meaningful, or not. We admit that there are still a few situations in which we identify deviations from monotonic trends now only in a few assessments. It was reassuring to us to see that most assessments had monotonic trends. However, for the few that did not, we did not supply an explanation. In these cases, we pointed out these cases.

What is the origin of the peculiar shape of the 90% contour in Figures 1, 2 and 11? In particular, what is the origin of the two lobes in the X/Y vs Z projection? Shouldn't the shape be related to the 3D point-spread function of the illumination?

We would like to thank the reviewer for this comment. The origin of the 2 lobes in X/Y vs Z projection are part of the simulation of a realistic PSF (it is also shown and explained in figure S1 – now figure 1). We have also added now detailed explanations about the way we calculated. The shape is directly related to the 3D PSF. Note that these are only the low intensity peripheral parts of the PSF. Nevertheless, these lobes should exist, where the Gaussian PSF model is known to be a good approximation to describe the center part of the PSF. The side lobes take up a small yet non-negligible fraction of the overall PSF energy, which is why it is important to use a PSF model that represents there measurement realistically. This is especially important when describing molecular positions from burst photons using burst analyses.

It might have not been clear enough, so we have now added additional contour lines of the numerical PSF model in these figures. Additionally, we corrected a small in the python representations of the PSF contour lines. On top of that, now that we ran comparative simulations, with the numerical and the Gaussian PSF models, we show both PSF shapes in a new version of figure S1, showing that as it comes to their shape, aside from the peripheral lobes, the majority of the PSF looks like a centered distribution. Nevertheless, the numerical PSF is still asymmetric. We decided to leave both PSF models to be presented to the readers, with thorough explanations on the reasons for using both models. Overall, the majority of our conclusions did not change due to the choice of the PSF model.

The labels and legends in the figures should be larger, to be more easily legible (at least such as in Fig. 3). It would help if all the quantitites (such as the 90% contour) were described in the caption instead in the text only.

Now the labels are larger, clearer (hopefully) and we added more contour plots describing the overlay of the PSF. We also did our best to describe well all presented information, including in dense figures that include many panels showing the results from many conditions and parameters. We hope that reviewer No. 1 will see the differences from the previous format of the manuscript, including the fact that it now includes much more simulations and parameters, following the requests of the other reviewers.

page 17, line 526: sentence 'However, the parameters that when....' seems impossible to understand.

The text has been revised over all of the manuscript, including this part.

page 17, line 540: multisport SMS

Corrected

page 17, line 546 and further: motional narrowing has a fixed, different meaning in traditional spectroscopy, recommend rephrasing

We wish to thank the reviewer for this comment. We do not discuss this effect anymore. The simulations with the longer durations were useful in showing that it was merely a question of the amount of information we acquired in the low diffusion coefficient simulations.

Reviewer 2 Report

              In this paper, the authors conducted simulation of photon signals generated by fluorescence burst detection-type single-molecules Förster resonance energy transfer (smFRET) measurements, including processes of diffusion of sample fluorescent molecules and fluorescence photon emission, using freely distributed software PyBroMo.  The results were evaluated mainly in two aspects. The first was the dispersion of the molecular position, at which the sample molecule emitted a photon that was judged to be included in a burst, near the laser focus.  The second was impure bursts, which were defined as bursts including photons emitted from other molecules than main target molecules, with their amount and the level of burst impurity, which is fraction of impure photons in a burst.   The authors first investigated those dependence on parameters for burst detection algorithm, used in freely distributed software FRETbursts, and concluded that the photon detection rate threshold F, was the most effective to reduce the position dispersion, amount of impure bursts and the level of burst impurity.  The authors also investigated the dependence on two experimental conditions, concentration and diffusion coefficient of the sample molecule, and concluded that lower concentration yields better results and that slower diffusion reduces influence of impure bursts while position dispersion was not improved.

              Because it is difficult to understand signal generation process of fluorescence burst measurements only from experimental results, the results of this manuscript would help smFRET experimentalists to understand its mechanism.  Simulations and analyses were well conducted and the conclusions were almost convincing.  As conclusion, I would recommend publication of this manuscript in Molecules after the following points are addressed.

Major concern:

l  Throughout the manuscript, the authors seem to claim that the smaller molecular position dispersion is better.  However, pimal goal of this kind of measurements must be accuracy of FRET measurements (e.g. average, distribution), and the positions, where detected photons were emitted, should not be a problem.  Since impure bursts degrade quality of burst signals, it is reasonable to reduce influence of impure bursts by reducing position dispersion.  However, even in the case of slow diffusion, in which impure bursts were suppressed, the authors seemed to try to reduce position dispersion or avoid formation of two sub-populations.  The authors should explain the reason why the position dispersion must be smaller, except for the effect of reducing impure bursts.

 Minor concerns:

l  Fig.1, 2, 11, S2:  In x-y plots, scaling in x- and y-axes should be same, like Fig.S1.  The distributions should be circular, but they look like ellipsoidal.

l  Fig.1, 2, 11, S2:  The meaning of colors of points is unclear.  Does the notation "burst number" in the figure captions mean that a single point can include more than one burst and the color represents its quantity?  Or does it mean the index of bursts?  If so, the figure is misleading because it seems like molecular position depends on the burst index, i.e. there are no red and green points and blue points gather near the center.

l  Fig.3:  Label or legend should be added to represent what each graph stands for.

l  Fig.4:  It is not reasonable in principle that the results along x- and y-coordinates were different.  The authors should mention the cause of the difference, its reproducibility and show error bars in graphs.  Are the similar differences seen in the result of Fig.8 and 10, while y-results are not shown in those figures?  Does the PSF shown in Fig.S1 have asymmetry to produce the difference?  If the difference is caused just by stochasticity of simulation, lateral position may be evaluated with r instead of x and y.

l  Fig.5, 6:  The last sentence in the figure caption should be deleted.

l  ll.303–304 in p.10:  The meaning of "increasing the minimal photon rate (F) allows better rejection of such instances." is unclear.  If the main molecule is detected as a burst with high photon rates, another molecule behind it can be included in the burst.  Its probability does not seem to depend on F value.  Explain more clearly.

l  Fig.6, 7:  m and F values are not shown.

l  Fig.8: Results along y-coordinate was not shown although it was mentioned in the figure caption.

l  Fig.8, 10:  m value was not specified for third panels from top.

l  Fig.1, 11:  What the "two sub-populations" means is unclear.  It should be clearly explained.  Does it mean two peaks in histograms above x-z and y-z plots?  If so, cross section or 1D projection along z-axis of PSF (Fig.S1) should be shown to show whether the PSF itself has such two peak-distribution or not.

l  The sentences may be improved for better readability.

Author Response

Reviewer No. 2

In this paper, the authors conducted simulation of photon signals generated by fluorescence burst detection-type single-molecules Förster resonance energy transfer (smFRET) measurements, including processes of diffusion of sample fluorescent molecules and fluorescence photon emission, using freely distributed software PyBroMo.  The results were evaluated mainly in two aspects. The first was the dispersion of the molecular position, at which the sample molecule emitted a photon that was judged to be included in a burst, near the laser focus.  The second was impure bursts, which were defined as bursts including photons emitted from other molecules than main target molecules, with their amount and the level of burst impurity, which is fraction of impure photons in a burst.   The authors first investigated those dependence on parameters for burst detection algorithm, used in freely distributed software FRETbursts, and concluded that the photon detection rate threshold F, was the most effective to reduce the position dispersion, amount of impure bursts and the level of burst impurity.  The authors also investigated the dependence on two experimental conditions, concentration and diffusion coefficient of the sample molecule, and concluded that lower concentration yields better results and that slower diffusion reduces influence of impure bursts while position dispersion was not improved.

              Because it is difficult to understand signal generation process of fluorescence burst measurements only from experimental results, the results of this manuscript would help smFRET experimentalists to understand its mechanism.  Simulations and analyses were well conducted and the conclusions were almost convincing.  As conclusion, I would recommend publication of this manuscript in Molecules after the following points are addressed.

We would like to express our sincere appreciation of the quite comprehensive report reviewer No. 2 provided. His review report has shown us that some of the content we provided in the previous version of the manuscript might have not been clear enough. Below, we provide our detailed response to reviewer No. 2, answering any question or doubt he has expressed.

As a general note: reviewer No. 2 said that this work deal with bursts of single-molecule FRET measurements. We would also like to mention as a general remark that the paper did not deal with photon bursts of single-molecule FRET as the main theme, but rather in general with photon bursts in single-molecule fluorescence detection, where single-molecule FRET is just one (very famous) application of it. We have referred to single-molecule FRET just as an example, in the end of the manuscript, serving as a proof-of-concept.

Major concern:

l  Throughout the manuscript, the authors seem to claim that the smaller molecular position dispersion is better.  However, pimal goal of this kind of measurements must be accuracy of FRET measurements (e.g. average, distribution), and the positions, where detected photons were emitted, should not be a problem. 

We would like to refer reviewer No. 2 to our main response, explaining we did not study single-molecule FRET in this work, but the more general analyses of single-molecule fluorescence bursts (where single-molecule FRET is a subset of such measurements, with bursts detected from two detection channels).

Since impure bursts degrade quality of burst signals, it is reasonable to reduce influence of impure bursts by reducing position dispersion.  However, even in the case of slow diffusion, in which impure bursts were suppressed, the authors seemed to try to reduce position dispersion or avoid formation of two sub-populations. 

We did not try to reduce any position dispersion. There is a difference between this claim and hypothesizing that burst impurity depends (among other parameters discussed in this manuscript) on the molecular position dispersion. The only place in the manuscript that might have hinted on the possible motivation we have to reduce the molecular position dispersion was in the Discussion chapter, where we explained that aside from reducing the concentration, another way to reduce burst impurity was by reducing the molecular position dispersion, (by increasing the values of the F burst search parameter), to decrease the probability of a second molecule to emit photons that will pass the burst search test.

In fact, we argue that reduced molecular dispersion via increasing the value of the F burst search parameter, will not degrade single molecule fluorescence signals much (and it is also shown in the manuscript), because still, molecules positioned closer to the center of the effective detection volume will emit photons at a high rate, which will be discovered by such burst search parameter values.

The authors should explain the reason why the position dispersion must be smaller, except for the effect of reducing impure bursts.

We understand this request and appreciate the reviewer’s critical ideas. We would like to note that the characteristics of impure bursts was our main focus in this paper. To emphasize this, we also modified the title of the manuscript, as well as the flow of the content in the text. Aside from the focus of this specific paper, we do not seem to find another specific reason for reducing the molecular position dispersion, other than being able to retrieve bursts with minimal impurity, even if the concentration of the measured molecules is close to the upper limit for these kind of measurements. We explained this point in the Discussion chapter (which is now explained in more details) – we explicitly explained that fighting burst impurity with measurements of molecules at even lower concentrations is a good solution, if acquisition time is not an issue.

Minor concerns:

l  Fig.1, 2, 11, S2:  In x-y plots, scaling in x- and y-axes should be same, like Fig.S1.  The distributions should be circular, but they look like ellipsoidal.

We would first like to note that the x & y scales in the x-y spreads were identical (-1 to 1) & (-1 to 1) – the width & height of these panels were not, which make the PSF contour look elliptical. We would like to note that our goal in the representation of z-x, z-y and x-y spreads shown as panels, one on top of the other, was also to allow the reader to visually compare the spreads in the different panels. Keeping width and height of the x-y spread panel would make it in a subplot, would force the height of the z-x & z-y spread panels to also change, which will make the asymmetric spread in z-x & z-y seem odd. The scales of the x & y coordinates in x-y spreads is identical and we wish to keep these figures as is, regarding the width & height of the panels.

I would like to mention that reviewer No. 3 also commented on the importance of properly showing the PSF shapes in the main text instead of the supplementary materials. Therefore, we moved figure S1 to the main text (it is now figure 1) to emphasize the size of shape of the PSF models used in this study.

l  Fig.1, 2, 11, S2:  The meaning of colors of points is unclear.  Does the notation "burst number" in the figure captions mean that a single point can include more than one burst and the color represents its quantity?  Or does it mean the index of bursts?  If so, the figure is misleading because it seems like molecular position depends on the burst index, i.e. there are no red and green points and blue points gather near the center.

The latter suggestion of reviewer No. 2 is correct. Each point is a photon of a burst, and the colors are following indexes of burst number. The figure is being built in the Jupyter notebook directly from the data, and hence is built like layers, where each point is added on top of the previous ones. The sole reason for observing less of some colors near the center, is that the colors of the final bursts (green & blue) cover the previous bursts (red & yellow) – the coverage is more efficient towards the center of the PSF, because of a larger density of burst photons there.

To solve this problem, we reduced the dots’ size and added some transparency to them. However, this solution is not perfect, because the density of photons from positions close to the center of the PSF will always be high.

We believe that the 1D projected histograms of positions help understand better these figures. Additionally, we believe the figures are now improved in that sense.

l  Fig.3:  Label or legend should be added to represent what each graph stands for.

      We have now supplemented almost all figures with legends, captions, and additional visual markers, to help make figures clearer, even when the amount of presented information is large.

l  Fig.4:  It is not reasonable in principle that the results along x- and y-coordinates were different.  The authors should mention the cause of the difference, its reproducibility and show error bars in graphs.  Are the similar differences seen in the result of Fig.8 and 10, while y-results are not shown in those figures?  Does the PSF shown in Fig.S1 have asymmetry to produce the difference?  If the difference is caused just by stochasticity of simulation, lateral position may be evaluated with r instead of x and y.

We would like to thank reviewer No. 2 for this comment. We have now added error analyses and error bars for all reported quantities. For some quantities (for instance, for the molecular dispersion), the errors were so low, they become invisible in figures. Therefore, we also added supplementary tables summarizing all quantities from all simulations with their respective error ranges. Additionally, we re-checked all of the computational procedures in our Jupyter notebooks and found some minor bugs, we have now fixed. All figures of this work were tested and reproduced. The addition of error bars helped a lot in answering this question raised by the reviewer. This discrepancy between x- and y-coordinates is now solved – deviations between the x- and y-coordinate molecular position dispersions are now the same within error ranges.

l  Fig.5, 6:  The last sentence in the figure caption should be deleted.

      Done.

l  ll.303–304 in p.10:  The meaning of "increasing the minimal photon rate (F) allows better rejection of such instances." is unclear.  If the main molecule is detected as a burst with high photon rates, another molecule behind it can be included in the burst.  Its probability does not seem to depend on F value.  Explain more clearly.

We have now explained this point more clearly. We also added additional assessments that correlate the burst impurity levels with the molecular positions in general and in particular the ones coming from impure bursts. We explain that although two molecules may traverse the high photon rate regime of the effective detection volume (and hence produce high from both molecules), there are more cases in which impure bursts arise from the second molecule when it is traversing the peripheral parts. The level of burst impurity found from calculation over many molecules is correlated with the probability, and this measure (the level of impurity in bursts) decreases when increasing the value of F, hence when acquiring photons from molecules that traversed a smaller volume, closer to the center of the effective excitation volume.

l  Fig.6, 7:  m and F values are not shown.

 Now they are shown in any figure that shows comparisons of different burst analysis parameter values

l  Fig.8: Results along y-coordinate was not shown although it was mentioned in the figure caption.

Thanks for notifying us on that. In the current version of the manuscript we: 1) show the molecular position histograms of all three coordinates; 2) we report the values of the molecular position dispersion of all three coordinates, in the summarizing Table S2; and 3) we report the molecular position dispersion in the z and x coordinates, explaining that the y-coordinate values are the same in within error ranges (see Table S2).

l  Fig.8, 10:  m value was not specified for third panels from top.

 The third panel from the top in these figures does not report the trends as a function of the m value, but rather the values of the burst size threshold. The difference is explained in the introduction and referred to throughout the text.

l  Fig.1, 11:  What the "two sub-populations" means is unclear.  It should be clearly explained.  Does it mean two peaks in histograms above x-z and y-z plots?  If so, cross section or 1D projection along z-axis of PSF (Fig.S1) should be shown to show whether the PSF itself has such two peak-distribution or not.

We do not refer to this point anymore, as we performed longer simulations that have shown that this is merely an artifact of lower amount of information, in 60 second simulations of low diffusion coefficient values (yielding a low amount of bursts, but with bursts of larger sizes).

l  The sentences may be improved for better readability.

We did our best to improve the readability of the sentences. We hope that reviewer No. 2 will read the current version of the manuscript and find that it is in a better and improved state.

Reviewer 3 Report

In solution-based single-molecule experiments, freely diffusing molecules are measured using a confocal microscope as they traverse the femtoliter-sized observation volume. The first step of the analysis is hereby the separation of bursts of fluorescence arising from single-molecule events from background signal. The parameters of the burst search are often chosen arbitrarily and the extent to which they influence the experimental results is unknown.

In this manuscript, Hagai et al aim to provide a systematic assessment of the burst search parameters. Using simulation, they address the question of the spatial position of selected molecules within the observation volume, which is not possible to study experimentally, in dependence of the parameters used for the burst search, such as the minimum photon number, the count rate threshold and the burst duration.

In general, the results conform to the expectations. Selection of bright bursts by more stringent criteria reduces the effective observation volume, as only molecules which pass the region of highest intensity are selected. More interestingly, the authors address the question of how to increase the “purity” of the measurement by reducing the contribution of multi-molecule events. If multiple populations are present, these multimolecule lead to the occurrence of events with average FRET efficiency and thus false results. To this end, the authors propose guidelines to minimize the impact of these artifacts by a choice of appropriate burst search parameters. While the topic of the manuscript will certainly be of interest to the single-molecule fluorescence community, I have a number of concerns both with respect to the procedures and the significance of the conclusions drawn.

The authors aimed to use a realistic model for the observation volume of the confocal microscope, which they generated using the PSFLab software. Generally, the point spread function in confocal microscopy is assumed to be approximately Gaussian shaped. This approximation is justified in the fact that the effective observation volume is given by the product of the excitation and detection point spread functions (PSF), convolved with the pinhole function (Webb RH (1996) Confocal optical microscopy. Reports on Progress in Physics 59(3):427–471; Hess ST, Webb WW (2002) Focal volume optics and experimental artifacts in confocal fluorescence correlation spectroscopy. Biophys J83(4):2300–2317). Additionally, in burst analysis the objective lens is often underfilled, resulting in a larger and more homogeneous observation volume. From my reading of the PSFLab software, it performs a calculation only of the illumination PSF, while not accounting for the detection by the same objective lens or the effects of the pinhole. This raises questions about the validity of the inhomogeneous PSF used in the simulations, which I suspect to overestimate the inhomogeneity of the observation volume. Specifically, in their study of slowly diffusing molecules, the authors discuss artifacts of multiple maxima in the spatial distributions of detected bursts, which are solely an effect of the inhomogeneous PSF. On my initial read, I was also confused by the shape of the point spread function, as the calculation is not mentioned explicitly in the text. I would thus strongly suggest that the simulations are repeated either with a true experimental PSF (which e.g. could be measured by a bead scan) or, more simply, using a “traditional” PSF model using a 3D Gaussian intensity distribution. This will provide a reference and allow to disentangle the effects of the inhomogeneous intensity distribution from the real effects of the burst search parameters.

Another major concern I have is the statistics of the reported results. Given that the data are simulated, it should be no major challenge to generate longer simulations than the short 60 second intervals chosen by the authors. While this is generally not a problem for the analysis of the spatial dispersion, the later conclusions regarding the purity and the analysis of the FRET efficiency histograms is limited by the available number of single-molecule events. It would thus be required to increase the statistics by performing longer simulations to verify the conclusions.

While I could follow the text, I also anticipate that it will be difficult for the general reader to not get lost in the forest of data and parameters. To this end, a clearer description of the applied burst search, supplemented by an illustration, might be helpful. While it is also appreciated that the authors aim to provide all data, as is also supported by their commendable efforts to lay the analysis workflow open by means of open-source software and Jupyter Notebook Python scripts, I think that the informational content of the figures could be reduced to the relevant sensitive parameters, while others could be moved to the SI.

 1.     In the simulations (Table 1), what is the assumed brightness of the molecules? Also, since the factor Fof the burst search determines the count rate threshold as a multiple of the background count rate, what is the background? I could find no reference for these numbers in the manuscript.

 2.     Since the authors use a complicated model for the PSF, this intensity distribution should be shown in the main text. On my initial read, I was left wondering how the asymmetric spatial distribution shown in Figure 1 could arise from the burst search. This should be made clearer. Also, what is the effective observation volume, i.e the size, of this PSF, in terms of the lateral and axial width used in FCS?

 3.     In Figure 1 (top), the spatial distribution in the absence of a minimum photon number is shown. However, in Figure 3, where the histograms are compared, only measurements with a minimum threshold of 10 photons per burst or more are shown. I was wondering what the dispersion would be for the case of no minimum photon number threshold. Specifically, the authors mention that a large number of events are detected outside of the observation volume for no minimum photon number threshold (lines 205-207), but did not include it in the comparison.

 4.     Then, what are the advantages of a narrow position dispersion? Is the accuracy of the measurement affected if excessive dispersion is present?

 5.     One of the parameters of the burst search is the minimum burst width, measured in milliseconds. This parameter surprisingly showed no influence on the spatial dispersion (Figure 4). The effect of this parameter should depend on the average diffusion time. What is the diffusion time in this case? I wonder if it would be better to express it as a ratio to the average diffusion time determined by FCS.

6.     Similarly, the probability to detect multiple molecules can be expressed as a function of the average number of particles in the volume (e.g. determined by FCS), as P(N>1) = 1-P(N=0)-P(N=1), where P is given by a Poisson distribution with mean. The purity level could be compared to this metric as a reference.

7.     The conclusions drawn from Figure 5 are limited by the statistics available and it is difficult to see the discussed trends in the noisy distributions.

8.     It is found that the purity of the bursts increases with increasing photon rate threshold F. I suspect that this would be an effect of overall shorter bursts due to the higher threshold and thus less chance of mixing, while low values of Fresult in longer bursts that have a higher likelihood to contain contributions from multiple molecules. Thus, it would be interesting to investigate the burst width distribution as a function of F.

9.     Instead of the concentration, would it be better to report the average number of molecules in the volume to provide a general parameter that is independent of the PSF size? Similarly, the diffusion time could be reported instead of the diffusion coefficient as a relative parameter that is independent of the shape and size of the PSF.

10.   For slow diffusion, a slight increase of the spatial dispersion is observed, which the authors attribute to the effect of motional narrowing. However, the instantaneous spatial position of every photon detection event is recorded, thus there is no averaging that could depend on the diffusion speed (e.g. the two position maxima are also observed in Figure 1). Rather, I would expect motional narrowing if the average position of the molecule for a given time interval were to be used. I am also unsure if the non-monotic change of the position dispersion as a function of the diffusion coefficient (Figure 10) is significant, given that there are no error bars given.

11.   In the recommendations provided from the study of the diffusion coefficient (lines 448-452), the authors suggest avoiding molecular position dispersion by choosing a large value for F.However, it is not clear why position dispersion is to be avoided and what it’s effect on the experimental result would be.

12.   The FRET efficiency distributions in Figure 12 are quite noisy, raising concerns with respect to the robustness of the applied Gaussian fitting. Given the data, I suspect that the result from the fitting is potentially biased by e.g. the choice of start parameters or the goodness-of-fit measure (i.e. reduced chi-sqare, MLE…), which is not mentioned in the text. These concerns could be minimized if better statistics were available for the histogram, to ensure that it is not just noise that results in the differences of the fit results for the histograms. While certainly an effect is expected, I suspect that it is exaggerated from the fit results.

13.   There exist filters to reduce the contributions of multimolecule events. One of these is the ALEX-2CDE filter, which is available in the FRETbursts software. Would this filter help to increase the purity of the bursts?

14.   Regarding the analysis of two FRET populations, I was wondering what the effect of differing brightness would be. Often, different FRET states are associated with different brightnesses, since the detection efficiency for the donor and acceptor are uneven. This results in a bias of the population fractions if stringent burst search criteria are applied since bright bursts are selected preferentially. The question of appropriate criteria for the burst search to provide accurate population weights in this case would be of high interest to the community.

Author Response

Reviewer No. 3

In solution-based single-molecule experiments, freely diffusing molecules are measured using a confocal microscope as they traverse the femtoliter-sized observation volume. The first step of the analysis is hereby the separation of bursts of fluorescence arising from single-molecule events from background signal. The parameters of the burst search are often chosen arbitrarily and the extent to which they influence the experimental results is unknown.

In this manuscript, Hagai et al aim to provide a systematic assessment of the burst search parameters. Using simulation, they address the question of the spatial position of selected molecules within the observation volume, which is not possible to study experimentally, in dependence of the parameters used for the burst search, such as the minimum photon number, the count rate threshold and the burst duration.

In general, the results conform to the expectations. Selection of bright bursts by more stringent criteria reduces the effective observation volume, as only molecules which pass the region of highest intensity are selected. More interestingly, the authors address the question of how to increase the “purity” of the measurement by reducing the contribution of multi-molecule events. If multiple populations are present, these multimolecule lead to the occurrence of events with average FRET efficiency and thus false results. To this end, the authors propose guidelines to minimize the impact of these artifacts by a choice of appropriate burst search parameters. While the topic of the manuscript will certainly be of interest to the single-molecule fluorescence community, I have a number of concerns both with respect to the procedures and the significance of the conclusions drawn.

We would like to extend our thanks to reviewer No. 3 for providing a thorough and comprehensive review report. We believe that his review was the major driving force in making this manuscript more complete.

The major requests made by reviewer 2 were:

·        Provide explanations for the c

·        Comparison of the simulations performed with the numerical PSF model, against simulations performed with a Gaussian-shaped PSF model

·        Comparisons of simulations and retrieved quantities between simulations of 60 seconds durations and simulations of longer durations.

·        Compare quantities achieved from ground-truth values in the simulation, with ones attained from FCS-like analyses

·        Improve figures and figure labels as well as data representation in general

We followed the reviewers’ suggestions and provided all the additional assessments asked by him. However, we do want to mention that adding these important estimations and verifications made the manuscript longer. We did the best we can to improve the readability of the manuscript, to make it more concise, and to keep a logical flow of the content, without losing on the comprehensiveness of the results reported in this work. We hope reviewer No. 2 can see all of the above when going over the manuscript in its current state and when comparing it to the previous version.

The authors aimed to use a realistic model for the observation volume of the confocal microscope, which they generated using the PSFLab software. Generally, the point spread function in confocal microscopy is assumed to be approximately Gaussian shaped. This approximation is justified in the fact that the effective observation volume is given by the product of the excitation and detection point spread functions (PSF), convolved with the pinhole function (Webb RH (1996) Confocal optical microscopy. Reports on Progress in Physics 59(3):427–471; Hess ST, Webb WW (2002) Focal volume optics and experimental artifacts in confocal fluorescence correlation spectroscopy. Biophys J83(4):2300–2317). Additionally, in burst analysis the objective lens is often underfilled, resulting in a larger and more homogeneous observation volume. From my reading of the PSFLab software, it performs a calculation only of the illumination PSF, while not accounting for the detection by the same objective lens or the effects of the pinhole. This raises questions about the validity of the inhomogeneous PSF used in the simulations, which I suspect to overestimate the inhomogeneity of the observation volume. Specifically, in their study of slowly diffusing molecules, the authors discuss artifacts of multiple maxima in the spatial distributions of detected bursts, which are solely an effect of the inhomogeneous PSF. On my initial read, I was also confused by the shape of the point spread function, as the calculation is not mentioned explicitly in the text. I would thus strongly suggest that the simulations are repeated either with a true experimental PSF (which e.g. could be measured by a bead scan) or, more simply, using a “traditional” PSF model using a 3D Gaussian intensity distribution. This will provide a reference and allow to disentangle the effects of the inhomogeneous intensity distribution from the real effects of the burst search parameters.

We wish to thank Reviewer No. 3 for this comment. We approached this comment in several directions, as follows:

·        Regarding the calculation of the numerical PSF model using PSFlab, the reviewer is absolutely right. PSFlab produces just the excitation PSF. Nevertheless, the PyBroMo simulation takes the multiplication of the PSF model produced by PSFlab, squared, assuming the detection PSF has a shape similar to the excitation PSF. Additionally, the simulation does not convolve the squared excitation PSF with the pinhole profile, as in most of the measurements of single-molecule fluorescence detection we have performed, we have not underfilled the back-aperture of the PSF, but rather over-filled it (the beam is expanded and collimated before entering into the back aperture of the objective lens. With overfilling the back-aperture, the convolution with the pinhole profile becomes irrelevant. We wish to thank the reviewer about his comment, as this helped us correct the text and properly present the procedure of the PSF treatment, which was lacking in the previous version of the manuscript. The text now extensively explain this point, in the Materials and Methods chapter.

·        We have now introduced all simulations, also with the Gaussian-shaped PSF. Over all the manuscript, we provide details both on the results of the simulations using the numerical PSF model and also the Gaussian PSF model. However, we chose to use a Gaussian PSF model that is comparable to the numerical PSF model we calculated, in its energy rather than in its FWHM. Owing to the fact that the numerical PSF model has side lobes, the central part of the PSF is narrower in the z coordinate. We chose to do so, so to mimic a comparison of two PSF models keeping the illumination power the same. The relative trends we report using the two PSF models, are similar in most cases (and when they are not, we comment on it in the text), but the absolute values are different in the two PSF models, owing to the narrower PSF when using the numerical PSF model, relative to the Gaussian model. Nevertheless, the most important message is that regardless of how one describes the PSF model, the general relative trends reported in this paper are kept similar in most cases – this is true both for the molecular position dispersion and for the burst impurity.

·        Regarding the peculiar molecular position histograms in low diffusion coefficient values, we took the reviewer’s recommendations and tested these simulations against same simulations with longer durations (180 seconds instead of 60 seconds, and not more, due to the huge file sizes of the diffusion trajectory files, produced by PyBroMo). The low diffusion simulations produce less bursts (that are larger in their size, though). Given the limited duration, we might have not collected enough bursts in these simulations. Indeed, the 180 s simulations have shown molecular position histograms with improved shape. In that context, we totally removed the whole section discussing the possible ‘motional narrowing’ and are grateful for reviewer No. 3 for this comment and his criticism, in general. 

Another major concern I have is the statistics of the reported results. Given that the data are simulated, it should be no major challenge to generate longer simulations than the short 60 second intervals chosen by the authors. While this is generally not a problem for the analysis of the spatial dispersion, the later conclusions regarding the purity and the analysis of the FRET efficiency histograms is limited by the available number of single-molecule events. It would thus be required to increase the statistics by performing longer simulations to verify the conclusions.

In regards to the statistics and the inferences made in this manuscript, we have now assessed all parameters and quantities, together with their corresponding error ranges (described thoroughly in the Materials and Methods chapter, as well as in all figure legends and tables.

As reviewer No. 3 suggested, there were situations in which quantities had non-negligible (yet small) error values. However, there were situations in which the error estimates were extremely small (such as in the calculation of the molecular position dispersion as the standard deviation of all molecular positions, owing to the vast amount of photons in these simulations (in most conditions, except for when using the size threshold of 80 photons per burst). Therefore, on top of showing all quantities in graphs with their corresponding error ranges, we have also provided supplementary tables that concentration all the values of all the quantities in this work, for all simulation conditions tested, including their error values.

Additionally, and as mentioned in the previous response to reviewer No. 3, we performed some of the simulations in durations of both in 60 & 180 seconds. We added a supplementary figure comparing all retrieved quantities in all burst analysis criteria tested in this paper, and show that the values of the quantities are , to the most part, the same within error ranges. All of these additions to the text are referred to wherever it was relevant. Additionally, each reported quantity in this paper has a corresponding figure produced by the Jupyter notebooks we deposited in Zenodo. However we presented these figures only for 1 (and sometimes 2) different simulation conditions, due to limited amount of figures. We have therefore also deposited all the raw figures of all the quantities for all simulations in Zenodo, for the readers to be able to inspect our results on all simulation conditions, rather than just follow the values in the supplementary tables as well as in the summarizing figures.

While I could follow the text, I also anticipate that it will be difficult for the general reader to not get lost in the forest of data and parameters. To this end, a clearer description of the applied burst search, supplemented by an illustration, might be helpful. While it is also appreciated that the authors aim to provide all data, as is also supported by their commendable efforts to lay the analysis workflow open by means of open-source software and Jupyter Notebook Python scripts, I think that the informational content of the figures could be reduced to the relevant sensitive parameters, while others could be moved to the SI.

We would like to thank the reviewer for this comment. At this point, and especially due to the enormous amount of additional information the reviewer asked us to add, to make this manuscript more complete, we decided not to graphically describe the burst analysis procedure. Partly because it was already shown many times in the literature, and also because we wanted to reduce the content in the main text, rather than add another figure.

Instead, we invested in reformatting the text and content of the manuscript, trying to reach a more logical flow of the text, we unified many figures in the main text, moved many to the supplementary materials, moved intermediate conclusions and discussions from the results chapter to the discussion chapter, and rewrote some of the introduction. However, we did so with caution not to misrepresent any parameter we thought is important for this work. That point should be emphasized in light of the additions to this manuscript, asked by the reviewer that should be referred to from the main text.

1.     In the simulations (Table 1), what is the assumed brightness of the molecules? Also, since the factor Fof the burst search determines the count rate threshold as a multiple of the background count rate, what is the background? I could find no reference for these numbers in the manuscript.

Both the molecular brightness (the simulated photon rate when a molecule is at the center of the effective detection volume), and the BG rates are parameter values we did not change, since we did not study them in this work. This is why we did not report them in table 1 (now Table S1). In the current version of the manuscript, we the simulated molecular brightness in the Materials and Methods, and both the simulated BG rates and the ones found in the analysis of the simulation, both in the Materials and Methods chapter, and in the supplementary tables. This point was actually important and we are sorry we did not mention it in the previous version of the manuscript – hence we would like (again) to thank reviewer No. 3. While the simulated BG rate represents the TRUE background (without the addition of fluorescence photons from molecules out-of-focus), the BG rate found by the analysis of the simulation results always has a value larger than the simulated one, and this difference is larger the larger the concentration is, hence the larger the amount of out-of-focus molecules that emitted a detected photon. The text now reports also this trend, as an important feature, however it is certainly not the main focus of this work. In the discussion, we describe a list of other parameters that might be important to systematically study by simulations, in future works.

 2.     Since the authors use a complicated model for the PSF, this intensity distribution should be shown in the main text. On my initial read, I was left wondering how the asymmetric spatial distribution shown in Figure 1 could arise from the burst search. This should be made clearer. Also, what is the effective observation volume, i.e the size, of this PSF, in terms of the lateral and axial width used in FCS?

We have moved the figure showing the PSF shapes (both the numerical & Gaussian PSF models), from the supplementary materials, to the main text.

3.     In Figure 1 (top), the spatial distribution in the absence of a minimum photon number is shown. However, in Figure 3, where the histograms are compared, only measurements with a minimum threshold of 10 photons per burst or more are shown. I was wondering what the dispersion would be for the case of no minimum photon number threshold. Specifically, the authors mention that a large number of events are detected outside of the observation volume for no minimum photon number threshold (lines 205-207), but did not include it in the comparison.

We thank reviewer No. 3 for pointing this out to us. Previously, the molecular position histograms projected on the y coordinate, were not shown in figure 3 (now figure S2). Now the molecular position histograms in z, x & y coordinates are shown. Then, the standard deviation of all positions per coordinate (the molecular position dispersion) in the x & y coordinates are the same within error ranges. When summarizing the molecular position dispersion, we do not report the value in the y coordinate, but state that the values are the same within the error rages. We also provide all values in the supplementary tables. There are now additional figures correlating certain quantities with the molecular position dispersion only in the z coordinate, mostly due to lack of space and too much information to show, already for one coordinate, when the trends both in the z and x/y coordinates are relatively similar. Now that we have all values reported in the supplementary tables, we were able to provide graphical representation of trends for some of the quantities, without loss of generality.

4.     Then, what are the advantages of a narrow position dispersion? Is the accuracy of the measurement affected if excessive dispersion is present?

The manuscript in its current state includes additional information showing the correlation between burst impurity and molecular position dispersion, if the values of F are varied (figure 6). Additionally, we also show that the shape of the molecular position dispersion coming from photons that originated from molecules that were not the main ones in impure bursts, have non-negligible contributions from positions on the periphery of the effective excitation volume, more than from its center, and that reducing the molecular position dispersion by increasing the value of F, leads to a reduction in burst impurity (figures 7 & S9-S13). Additionally, we added additional explanations and discussions in the discussion chapter, to explain the relation between burst impurity and molecular position dispersion. We also understood that this is basically our main focus in this paper, and hence modified the title of this work. The details added to the manuscript made it now more convincing that reducing the molecular position dispersion leads to reduction in the burst impurity.

The rationalization behind it (also given in the text), is that the smaller the volume is about the center of the PSF from which photons are identified, the lower the amount of photons arising from molecules other than the main ones, because most of the main molecules produce bursts of photons at high rates when close to the center of the PSF, while most of the ‘other’ molecules are not necessarily found at the same position in the same time, and hence produce photons at lower rates (which is also the connection to the choice of the F threshold).

5.     One of the parameters of the burst search is the minimum burst width, measured in milliseconds. This parameter surprisingly showed no influence on the spatial dispersion (Figure 4). The effect of this parameter should depend on the average diffusion time. What is the diffusion time in this case? I wonder if it would be better to express it as a ratio to the average diffusion time determined by FCS.

There was an effect of the choice of the burst width threshold on the position dispersion, but very small. We think this is because although burst width and size are related, they are not correlated 1-to-1. In fact, scatter plots of burst widths vs. sizes show that there is quite a distribution of possibilities, such as larger sizes with seemingly no changes in width, or vice versa. That is because there are many bursts with a high variability in the different instantaneous photon rates, along bursts. Since we assessed the burst width threshold on bursts that were acquired with an F value of 6, it may be interesting to test it against burst that were acquired using a higher F value. However, the higher the F values are, the shorter the bursts are (as can also be seen in the burst width histograms in the second panel from the top of figure S18). Additionally, figure S18 teaches us that using different burst search parameter values, m & F, leads to bursts with different widths, which makes it irrelevant to test the burst width threshold against constant subset of its values (0.5 & 1.0 ms, in our case).

Regarding retrieving the diffusion time from FCS, I must say I also tried (see the values in the supplementary table S2, as well as in figures S14 & S17, especially S17), and the retrieved values are too inaccurate to be used. Mean burst width, were the most useful in this sense (Figures S18 & S19).   

6.     Similarly, the probability to detect multiple molecules can be expressed as a function of the average number of particles in the volume (e.g. determined by FCS), as P(N>1) = 1-P(N=0)-P(N=1), where P is given by a Poisson distribution with mean. The purity level could be compared to this metric as a reference.

We would like to thank reviewer No. 3 for this important comment. We went the extra miles to provide a comparison of our estimates of burst impurities, against estimates from Poisson statistics relying on FCS-derived. We added photon timestamp autocorrelation calculations to our Jupyter notebooks, using the Pycorrelate code. By doing so, we were able to calculate photon timestamp autocorrelations of photon bursts. Then we fitted all curves in all cases, with a regular 3D diffusion model, usually used in FCS. Then, after retrieving the values of(that unlike the diffusion times, had plausible fitting error ranges), we calculated P(N>1), but in a different way. By definition, P(N=0) for photons of identified bursts do not have a clear meaning, because in these regions of the measurements, in side bursts, there is at least 1 molecule. Therefore we provided an approximate calculation, with the expression suggested by reviewer No. 3, normalized to 1 – P(N=0),

PBurst(N>1)=[1 – P(N=0) – P(N=1)] / [1 – P(N=0)]

Then we used this estimate as an alternative estimation of burst impurity level. In some cases (most of them), the trends reported with this measure were relatively comparable to the trends reported using the simulation-driven quantities for impurity. We reported all cases. We found it important enough to cherish this alternative measure an additional sub-section in the results. However, we did express our refrain from using this approach, both in this sub-section, in the discussion chapter and in the supplementary materials.

1.      First, the quantities of burst impurity (the fraction of impure bursts, the mean impurity in bursts, and a newly added quantity – the fraction of impure photons over all photons) are an outcome of our exact knowledge of the ground truth for each identified photon in bursts. The quantities derived from FCS, go over additional steps of calculation of an autocorrelation function, fitting it, and using Poisson statistics with best fit values. That’s quite a long road to produce a quantity that is directly achieved from the ground-truth of the simulation.

2.      FCS is perfect when performing measurements of molecules in concentrations that yield ~1 molecule in the effective excitation volume, at any given moment! This produces a signal with a clear mean, and temporal fluctuations around it. In confocal-based SMFD,<100 pM concentrations are used, yielding not 1, but 2 Poisson processes, background and signal (as bursts), with two different photon rates, which is quite different than in classical FCS measurements. Now, although the BG should in principle not be correlated in time, still the description of FCS deviates for measurements of bursts. Additionally, many before have shown how the quantification of the concentration (from) in this regime, becomes inaccurate.

3.      As is shown in this manuscript for burst photons, their molecular positions constitute different volumes as a function of the burst analysis parameter values. Therefore quantities such asdepend on changing effective volume.

4.      But also, the shape of the molecular position histograms deviate from the Gaussian approximation. Even in simulations using the Gaussian-shaped PSF, the shaped of the molecular position histograms after different burst analyses criteria, change. This puts in question, the use of the classical model used for fitting FCS curves to a model.

5.      Our modified calculation of PBurst(N>1), is also an approximation, because bursts include also BG photons, hence it becomes hard to decide how to calculate P(N>1) from burst photon timestamp autocorrelation curves.

Having said all of the above, it seems that PBurst(N>1) can serve as a nice alternative for the estimation of burst impurity, but only relatively, since its absolute values deviated much relative to quantities that were calculated directly from the ground-truth of burst impurity in the simulation.

7.     The conclusions drawn from Figure 5 are limited by the statistics available and it is difficult to see the discussed trends in the noisy distributions.

Reviewer No. 3 is absolutely right, especially when it comes to high burst size threshold values. Figure 5 (now figure S5) had vertical lines showing the mean impurity levels on top of the burst impurity histograms. We have now added error ranges (for all quantities reported), not only to the supplementary table S2, but also directly to figure S5, as well as the amount of bursts in each burst analysis criteria (each panel). Therefore, anybody can read this figure and judge which values are statistically meaningful and which are not. Additionally, it is worth mentioning that this figure was for the simulation with a duration of 60 s. We performed the same simulation for a duration of 180 s and reported this quantity (as well as all other) to be the same within error ranges per given burst analysis conditions, and that the quantities retrieved from the 180 s simulation simply had smaller error ranges (Figure S21).

8.     It is found that the purity of the bursts increases with increasing photon rate threshold F. I suspect that this would be an effect of overall shorter bursts due to the higher threshold and thus less chance of mixing, while low values of Fresult in longer bursts that have a higher likelihood to contain contributions from multiple molecules. Thus, it would be interesting to investigate the burst width distribution as a function of F.

We thank reviewer No. 3 for this comment. Indeed this was an additional parameter we tested, and the results are summarized in the text, in the tables and in figures S18 & S19. The reviewer is right, of course, but it was better to test (systematically assess) than just assume. Therefore (again) we thank the reviewer for pointing towards this and helping enhance the strength of this work.

9.     Instead of the concentration, would it be better to report the average number of molecules in the volume to provide a general parameter that is independent of the PSF size? Similarly, the diffusion time could be reported instead of the diffusion coefficient as a relative parameter that is independent of the shape and size of the PSF.

We wanted to report the input parameters we inserted into the simulation and throughout the manuscript, stay coherent and refer to simulations as they were defined. Additionally, we provided a detailed reference to these FCS-like quantities both in the text and in our response. The use of FCS-driven parameters in low concentration (<100 pM regime) us interesting, but aside from the side focus we gave it in this work, we believe it is beyond the scope of this work, owing to the reasons I detailed above.

10.   For slow diffusion, a slight increase of the spatial dispersion is observed, which the authors attribute to the effect of motional narrowing. However, the instantaneous spatial position of every photon detection event is recorded, thus there is no averaging that could depend on the diffusion speed (e.g. the two position maxima are also observed in Figure 1). Rather, I would expect motional narrowing if the average position of the molecule for a given time interval were to be used. I am also unsure if the non-monotic change of the position dispersion as a function of the diffusion coefficient (Figure 10) is significant, given that there are no error bars given.

As I mentioned before, this is clearly an artifact of low number of bursts in a simulation. We have seen how these simulations, lasting 180 seconds, provide 'better looking’ molecular position histograms. Therefore, we removed the whole discussion on ‘motional narrowing’, which has turned out to be far-fetched, with the help of the excellent review reports given here.

11.   In the recommendations provided from the study of the diffusion coefficient (lines 448-452), the authors suggest avoiding molecular position dispersion by choosing a large value for F.However, it is not clear why position dispersion is to be avoided and what it’s effect on the experimental result would be.

See my response to reviewer No. 2 as well my response to reviewer No. 3 answering point No. 4.  

12.   The FRET efficiency distributions in Figure 12 are quite noisy, raising concerns with respect to the robustness of the applied Gaussian fitting. Given the data, I suspect that the result from the fitting is potentially biased by e.g. the choice of start parameters or the goodness-of-fit measure (i.e. reduced chi-sqare, MLE…), which is not mentioned in the text. These concerns could be minimized if better statistics were available for the histogram, to ensure that it is not just noise that results in the differences of the fit results for the histograms. While certainly an effect is expected, I suspect that it is exaggerated from the fit results.

We would like to thank the reviewer for this comment. To answer this concern we did the following:

1.      We performed the fitting procedure on the results of 4 different simulations of the same conditions – 2 PSF models and 2 simulation durations (60 & 180 seconds). The longer simulation durations yielded more bursts, hence cleaner histograms, however the results (as reported in Table S3), as it comes to the trend in the deviation between the expected and best fit mean FRET efficiency values as a function of increasing F value were indeed sometimes different than what we expected. This is why we moved to the next step

2.      Table S3 shows a clear correlation between changes in the mean FRET efficiency values and the populations’ fraction. Therefore, we performed the same fits, only with the value of the populations’ fraction fixed to its ground-truth value, to see whether we can retrieve the expected trend in the inaccuracy of the FRET efficiency values. Table S3 reports also these results, and indeed, the higher the value of F was, the smaller the deviation was between the expected and retrieved values of the FRET efficiencies, also in the longer simulations, within error ranges.

3.      Table S3 reports the best fit results now along with the fitting errors that are a result of the fitting procedure.

4.      We left the reports on the statistical measures of the fitting procedure in the Jupyter Notebooks that report any piece of detail, for the thorough reader, and mostly due to lack of space. We reported in the Materials and Methods the whole procedure used in the fitting the notebooks

5.      We added a paragraph in the Discussion explaining why burst impurity is only one out of other important parameters that could contribute to an inaccuracy in retrieving the mean FRET efficiency, so that it will be clear that we are not trying to exaggerate the results of the assessment of mean FRET efficiency inaccuracy. This paragraph also explains why although in common use, Gaussian fitting is not the optimal approach to retrieve mean FRET efficiencies. The paragraph also explained why comparisons of FRET histograms driven from varying F values, but not different burst size threshold values, was the best choice (statistical comparison is hard from fits to different datasets having very different total number of items)

6.      On top of testing the effect of burst impurity on the accuracy of the retrieved mean FRET efficiencies on simulated data, we added also an experimental single-molecule FRET control, testing this approach on a measurements of a mixture of two FRET constructs. The data was taken from a previous work of Lerner, Ingargiola & Weiss. This experimental example has also proven useful in proving this point.

13.   There exist filters to reduce the contributions of multimolecule events. One of these is the ALEX-2CDE filter, which is available in the FRETbursts software. Would this filter help to increase the purity of the bursts?

ALEX-2CDE, as well as dual channel burst search, are burst analysis tools useful in the case of ALEX measurements. ALEX measurements in the context of this manuscript, are a special case of confocal-based SMFD. I refer to it by citation, in the discussion chapter, when describing other burst search techniques and parameters not covered in this manuscript.

14.   Regarding the analysis of two FRET populations, I was wondering what the effect of differing brightness would be. Often, different FRET states are associated with different brightnesses, since the detection efficiency for the donor and acceptor are uneven. This results in a bias of the population fractions if stringent burst search criteria are applied since bright bursts are selected preferentially. The question of appropriate criteria for the burst search to provide accurate population weights in this case would be of high interest to the community.

We would like to thank the reviewer for raising this point. This is yet another parameter we did not focus on. We focused on the burst impurity mostly. We did refer to this question in the discussion chapter, when discussing things left to do in the future. In that context, what will be more interesting would be to specifically assess whether beta- and gamma-weighted burst search and selection, does the job it is supposed to do. In that context, whether or not (and to what extent), two subpopulations with two different brightnesses get balanced.

Round  2

Reviewer 3 Report

The revised manuscript is significantly improved both in content and readability. I would like to thank the authors for the impressive amount of additional work that has been performed based on my comments within the short timeframe of the revisions.